# Calcium Orthophosphate (CaPO$_4$)-Based Bioceramics: Preparation, Properties, and Applications

Sergey V. Dorozhkin

Kudrinskaja sq. 1-155, 123242 Moscow, Russia; sedorozhkin@yandex.ru

**Abstract:** Various types of materials have been traditionally used to restore damaged bones. In the late 1960s, a strong interest was raised in studying ceramics as potential bone grafts due to their biomechanical properties. A short time later, such synthetic biomaterials were called bioceramics. Bioceramics can be prepared from diverse inorganic substances, but this review is limited to calcium orthophosphate (CaPO$_4$)-based formulations only, due to its chemical similarity to mammalian bones and teeth. During the past 50 years, there have been a number of important achievements in this field. Namely, after the initial development of bioceramics that was just tolerated in the physiological environment, an emphasis was shifted towards the formulations able to form direct chemical bonds with the adjacent bones. Afterwards, by the structural and compositional controls, it became possible to choose whether the CaPO$_4$-based implants would remain biologically stable once incorporated into the skeletal structure or whether they would be resorbed over time. At the turn of the millennium, a new concept of regenerative bioceramics was developed, and such formulations became an integrated part of the tissue engineering approach. Now, CaPO$_4$-based scaffolds are designed to induce bone formation and vascularization. These scaffolds are usually porous and harbor various biomolecules and/or cells. Therefore, current biomedical applications of CaPO$_4$-based bioceramics include artificial bone grafts, bone augmentations, maxillofacial reconstruction, spinal fusion, and periodontal disease repairs, as well as bone fillers after tumor surgery. Prospective future applications comprise drug delivery and tissue engineering purposes because CaPO$_4$ appear to be promising carriers of growth factors, bioactive peptides, and various types of cells.

**Keywords:** calcium orthophosphates; hydroxyapatite; tricalcium phosphate; bioceramics; biomaterials; grafts; biomedical applications; tissue engineering

## 1. Introduction

One of the most exciting and rewarding areas of the engineering discipline involves development of various devices for healthcare. Some of them are implantable. Examples comprise sutures, catheters, heart valves, pacemakers, breast implants, fracture fixation plates, nails and screws in orthopedics, various filling formulations, orthodontic wires, total joint replacement prostheses, etc. However, in order to be accepted by the living body without any unwanted side effects, all implantable items must be prepared from a special class of tolerable materials, called biomedical materials or biomaterials, in short. The physical character of the majority of the available biomaterials is solids [1,2].

From the material point of view, all types of solids are divided into four major groups: metals, polymers, ceramics, and various blends thereof, called composites. Similarly, all types of solid biomaterials are also divided into the same groups: biometals, biopolymers, bioceramics, and biocomposites. All of them play very important roles in both replacement and regeneration of various human tissues; however, setting biometals, biopolymers, and biocomposites aside, this review is focused on bioceramics only. In general, bioceramics comprise various polycrystalline materials, amorphous materials (glasses), and blends thereof (glass-ceramics). Nevertheless, the chemical elements used to manufacture bioceramics form just a small set of the periodic table; namely, bioceramics might be prepared

from alumina, zirconia, magnesia, carbon, silica-contained, and calcium-contained compounds, as well as from a limited number of other compounds. All these compounds might be manufactured in both dense and porous forms in bulk, as well as in the forms of crystals, powders, particles, granules, scaffolds, and/or coatings [1–3].

As seen from the above, the entire subject of bioceramics is still rather broad. To specify it further, let me limit myself by a description of calcium orthophosphate (abbreviated as $CaPO_4$)-based formulations only. If compared with other types of bioceramics (such as alumina, zirconia, calcium silicates, calcium sulfate, etc.), the main feature and superiority of $CaPO_4$ is based on their chemical similarity to the composition of calcified tissues of mammals (bones, teeth, and deer antlers) and the need for versatile and risk-free bone substitute biomaterials immediately available without the constraint of bone grafts. One of the major properties of most types of $CaPO_4$ is their osteoconductivity, an ability to favor bone healing and to bind firmly to bone tissues. In addition, some types of $CaPO_4$ have been shown to be able to initiate bone formation de novo in nonosseous sites [1–3]. Therefore, $CaPO_4$ bioceramics are widely used in a number of different applications throughout the body, covering all areas of the skeleton. The examples include healing of bone defects, fracture treatment, total joint replacement, bone augmentation, orthopedics, cranio-maxillofacial reconstruction, spinal surgery, otolaryngology, ophthalmology, and percutaneous devices [1–3], as well as dental fillings and periodontal treatments [4]. Furthermore, they are also used in nonosseous applications, such as ocular implants, allowing eye movements. Depending upon the required properties, different types of $CaPO_4$ might be used. For example, Figure 1 displays some randomly chosen samples of the commercially available $CaPO_4$ bioceramics for bone graft applications. One should note that the global bone grafts and substitutes market was valued at USD 2.65 billion in 2020 and is projected to reach USD 3.36 billion by 2028, registering a cumulative annual growth rate of ~4.3% from 2021 to 2028 [5]. This clearly demonstrates the biomedical perspectives of $CaPO_4$-based bioceramics.

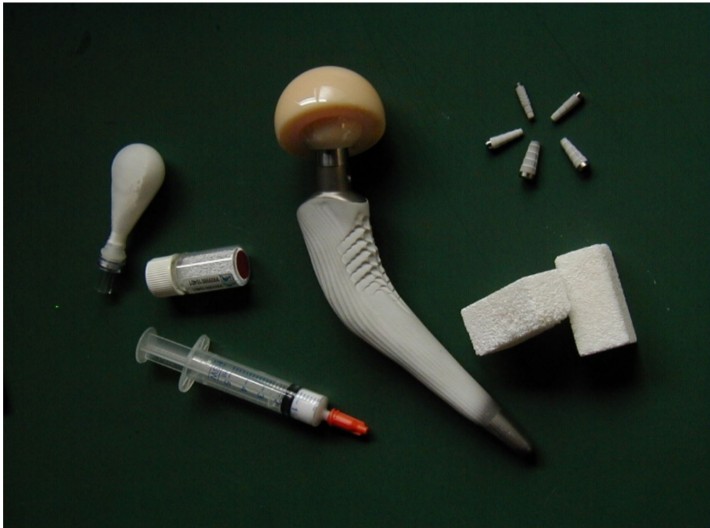

**Figure 1.** Several examples of the commercial $CaPO_4$-based bioceramics.

A list of the available $CaPO_4$, including their standard abbreviations and major properties, is summarized in Table 1 [3,6]. To narrow the subject further, with a few important exceptions, bioceramics prepared from undoped and unsubstituted $CaPO_4$ will be considered and discussed only. Due to this reason, $CaPO_4$-based bioceramics prepared from biological resources, such as bones, teeth, corals, antlers, etc. [7–14], including food [15] and animal wastes [16], as well as various types of ion-substituted $CaPO_4$ [17–41], including rhenanite $NaCaPO_4$ and chlorapatite $Ca_{10}(PO_4)_6Cl_2$, are not considered. The readers interested in both topics are advised to study the original publications.

**Table 1.** Existing calcium orthophosphates and their major properties [3,6].

| Ca/P Molar Ratio | Compounds and Their Typical Abbreviations | Chemical Formula | Solubility at 25 °C, -log($K_s$) | Solubility at 25 °C, g/L | pH Stability Range in Aqueous Solutions at 25 °C |
|---|---|---|---|---|---|
| 0.5 | Monocalcium phosphate monohydrate (MCPM) | $Ca(H_2PO_4)_2 \cdot H_2O$ | 1.14 | ~18 | 0.0–2.0 |
| 0.5 | Monocalcium phosphate anhydrous (MCPA or MCP) | $Ca(H_2PO_4)_2$ | 1.14 | ~17 | [c] |
| 1.0 | Dicalcium phosphate dihydrate (DCPD), mineral brushite | $CaHPO_4 \cdot 2H_2O$ | 6.59 | ~0.088 | 2.0–6.0 |
| 1.0 | Dicalcium phosphate anhydrous (DCPA or DCP), mineral monetite | $CaHPO_4$ | 6.90 | ~0.048 | [c] |
| 1.33 | Octacalcium phosphate (OCP) | $Ca_8(HPO_4)_2(PO_4)_4 \cdot 5H_2O$ | 96.6 | ~0.0081 | 5.5–7.0 |
| 1.5 | α-Tricalcium phosphate (α-TCP) | $\alpha\text{-}Ca_3(PO_4)_2$ | 25.5 | ~0.0025 | [a] |
| 1.5 | β-Tricalcium phosphate (β-TCP) | $\beta\text{-}Ca_3(PO_4)_2$ | 28.9 | ~0.0005 | [a] |
| 1.2–2.2 | Amorphous calcium phosphates (ACP) | $Ca_xH_y(PO_4)_z \cdot nH_2O$, $n = 3\text{–}4.5$; 15%–20% $H_2O$ | [b] | [b] | ~5–12 [d] |
| 1.5–1.67 | Calcium-deficient hydroxyapatite (CDHA or Ca-def HA) [e] | $Ca_{10-x}(HPO_4)_x(PO_4)_{6-x}(OH)_{2-x}$ $(0 < x < 1)$ | ~85 | ~0.0094 | 6.5–9.5 |
| 1.67 | Hydroxyapatite (HA, HAp, or OHAp) | $Ca_{10}(PO_4)_6(OH)_2$ | 116.8 | ~0.0003 | 9.5–12 |
| 1.67 | Fluorapatite (FA or FAp) | $Ca_{10}(PO_4)_6F_2$ | 120.0 | ~0.0002 | 7–12 |
| 1.67 | Oxyapatite (OA, OAp, or OXA) [f], mineral voelckerite | $Ca_{10}(PO_4)_6O$ | ~69 | ~0.087 | [a] |
| 2.0 | Tetracalcium phosphate (TTCP or TetCP), mineral hilgenstockite | $Ca_4(PO_4)_2O$ | 38–44 | ~0.0007 | [a] |

[a] These compounds cannot be precipitated from aqueous solutions. [b] Cannot be measured precisely. However, the following values were found: $25.7 \pm 0.1$ (pH = 7.40), $29.9 \pm 0.1$ (pH = 6.00), $32.7 \pm 0.1$ (pH = 5.28). The comparative extent of dissolution in acidic buffer is ACP >> α-TCP >> β-TCP > CDHA >> HA > FA. [c] Stable at temperatures above 100 °C. [d] Always metastable. [e] Occasionally, it is called "precipitated HA (PHA)". [f] Existence of OA remains questionable.

## 2. General Knowledge and Definitions

A number of definitions have been developed for the term "biomaterials". For example, by the end of the 20th century, the consensus developed by the experts was the following: biomaterials were defined as synthetic or natural materials to be used to replace parts of a living system or to function in intimate contact with living tissues [42]. In September 2009, a more advanced definition was introduced: "A biomaterial is a substance that has been engineered to take a form which, alone or as part of a complex system, is used to direct, by control of interactions with components of living systems, the course of any therapeutic or diagnostic procedure, in human or veterinary medicine" [43]. Further, in 2018, the term biomaterial was redefined as "a material designed to take a form that can direct, through interactions with living systems, the course of any therapeutic or diagnostic procedure" [44]. According to the Williams, "The two critical parts of this definition relate to the objectives of the systems in which a biomaterial is used and the fact that the material has to interact with living systems, in most cases parts of the human body, in order for these objectives to be realized. This definition, and indeed, the whole concept of biomaterials science, applies equally to situations involving implantable devices, artificial organs, tissue engineering templates, nonviral gene vectors, drug delivery systems and contrast agents" [45]. The

definition alterations were accompanied by a shift in both the conceptual ideas and the expectations of biological performance, which mutually changed in time. However, one should stress that any artificial materials that are in contact with skin, such as hearing aids and wearable artificial limbs, are not included in the definition of biomaterials since the skin acts as a protective barrier between the body and the external world [1,2].

In general, the biomaterials discipline is founded in the knowledge of the synergistic interaction of material science, biology, chemistry, medicine, and mechanical science and it requires the input of comprehension from all these areas so that potential implants perform adequately in a living body and interrupt normal body functions as little as possible [46]. As biomaterials deal with all aspects of the material synthesis and processing, the knowledge in chemistry, material science, and engineering appears to be essential. On the other hand, since clinical implantology is the main purpose of biomaterials, biomedical sciences become the key part of the research. These include cell and molecular biology, histology, anatomy, and physiology. The final aim is to achieve the correct biological interaction of the artificial grafts with living tissues of a host. Thus, to achieve the goals, several stages have to be performed, such as material synthesis, design, and manufacturing of prostheses, followed by various types of tests. Furthermore, before clinical applications, any potential biomaterial must also pass all regulatory requirements [47].

The major difference between biomaterials and other classes of materials lays in their ability to remain in a biological environment while neither damaging the surroundings nor being damaged in that process. Therefore, biomaterials must be distinguished from *biological materials* because the former are the materials that are accepted by living tissues and, therefore, they might be used for tissue replacements, while the latter are just the materials being produced by various biological systems (wood, cotton, bones, chitin, etc.) [48]. Furthermore, there are *biomimetic materials*, which are not made by living organisms but have composition, structure, and properties similar to those of biological materials. Concerning the subject of the current review, *bioceramics* (or biomedical ceramics) are defined as biomaterials having a ceramic origin. Now, it is important to define the meaning of ceramics. According to Wikipedia, the free encyclopedia: "The word *ceramic* comes from the Greek word κεραμικός (*keramikos*), "of pottery" or "for pottery", from κέραμος (*keramos*), "potter's clay, tile, pottery". The earliest known mention of the root "ceram-" is the Mycenaean Greek *ke-ra-me-we*, "workers of ceramics", written in Linear B syllabic script. The word "ceramic" may be used as an adjective to describe a material, product or process, or it may be used as a noun, either singular, or, more commonly, as the plural noun "ceramics". A ceramic material is an inorganic, nonmetallic, often crystalline oxide, nitride or carbide material. Some elements, such as carbon or silicon, may be considered ceramics. Ceramic materials are brittle, hard, strong in compression, weak in shearing and tension. They withstand chemical erosion that occurs in other materials subjected to acidic or caustic environments. Ceramics generally can withstand very high temperatures, such as temperatures that range from 1000 to 1600 °C (1800 to 3000 °F). Glass is often not considered a ceramic because of its amorphous (noncrystalline) character. However, glassmaking involves several steps of the ceramic process and its mechanical properties are similar to ceramic materials" [49]. Similar to any other type of biomaterials, bioceramics can have structural functions as joint or tissue replacements, and be used as coatings to improve the biocompatibility, as well as function as resorbable lattices, providing temporary structures and frameworks that are dissolved and/or replaced as the body rebuilds the damaged tissues [50–53]. Some types of bioceramics feature a drug-delivery capability [54–57].

In medicine, bioceramics are needed to alleviate pain and restore functions of diseased or damaged calcified tissues (bones and teeth) of the body. A great challenge facing its medical application is, first, to replace and, second, to regenerate old and deteriorating bones with a biomaterial that can be replaced by a new mature bone without transient loss of a mechanical support [1,2]. The excellent performance of the specially designed bioceramics that have survived these clinical conditions represents one of the most remarkable accomplishments of research, development, production, and quality assurance before the

end of the past century [50]. Regarding $CaPO_4$ bioceramics, a surface bioactivity appears to be its major feature. It contributes to a bone bonding ability and enhances new bone formation [58].

### 3. Bioceramics of $CaPO_4$

#### *3.1. History*

The detailed history of HA and other types of $CaPO_4$, including the subject of $CaPO_4$ bioceramics, as well as description of their past biomedical applications, might be found elsewhere [59,60], where the interested readers are referred. One should just note that the earliest book devoted to $CaPO_4$ bioceramics was published in 1983 [61].

#### *3.2. Chemical Composition and Preparation*

Currently, $CaPO_4$ bioceramics can be prepared from various sources [7–16]. Nevertheless, up to now, all attempts to synthesize bone replacement materials for clinical applications featuring the physiological tolerance, biocompatibility, and a long-term stability have had only relative success; this clearly demonstrates both the superiority and a complexity of the natural structures [62].

In general, a characterization of $CaPO_4$ bioceramics should be performed from various viewpoints such as the chemical composition (including stoichiometry and purity), homogeneity, phase distribution, morphology, grain sizes and shape, grain boundaries, crystallite size, crystallinity, pores, cracks, surface roughness, etc. From the chemical point of view, the vast majority of $CaPO_4$ bioceramics are based on HA [63–67], both types of TCP [68–78], and various multiphasic formulations thereof [79]. Biphasic formulations (commonly abbreviated as BCP–biphasic calcium phosphate) are the simplest among the latter ones. They include β-TCP + HA [80–88], α-TCP + HA [89–91], and biphasic TCP (commonly abbreviated as BTCP), consisting of α-TCP and β-TCP [92–97]. In addition, triphasic formulations (HA + α-TCP + β-TCP) have been prepared as well [98–101]. Further details on this topic can be found in a special review [79]. Leaving aside a big subject of DCPD-forming self-setting formulations [102,103], one should note that just a few publications on bioceramics prepared from other types of $CaPO_4$ are available [104–112].

The preparation techniques of various types of $CaPO_4$ have been extensively reviewed in the literature [6,113–117], where the interested readers are referred. Briefly, when compared to both α- and β-TCP, HA is a more stable phase under the physiological conditions, as it has a lower solubility (Table 1) and, thus, slower resorption kinetics [118–120]. Therefore, the BCP concept is determined by the optimum balance of a more stable phase of HA and a more soluble TCP. Due to a higher biodegradability of the α- or β-TCP component, the reactivity of BCP increases with the TCP/HA ratio increasing. Thus, in vivo bioresorbability of BCP can be controlled through the phase composition [81]. Similar conclusions are also valid for the biphasic TCP (in which α-TCP is a more soluble phase), as well as for both triphasic (HA, α-TCP, and β-TCP) and yet more complex formulations [79].

As implants made of sintered HA are found in bone defects for many years after implantation (Figure 2, bottom), bioceramics made of more soluble types of $CaPO_4$ [68–112,121,122] are preferable for the biomedical purposes (Figure 2, top). Furthermore, the experimental results showed that BCP had a higher ability to adsorb fibrinogen, insulin, or type I collagen than HA [123]. Thus, according to both observed and measured bone formation parameters, $CaPO_4$ bioceramics have been ranked as follows: low sintering temperature BCP (rough and smooth) ≈ medium sintering temperature BCP ≈ TCP > calcined low sintering temperature HA > non-calcined low sintering temperature HA > high sintering temperature BCP (rough and smooth) > high sintering temperature HA [124]. This sequence was developed in the year 2000 and, thus, neither multiphase formulations nor other $CaPO_4$ are included.

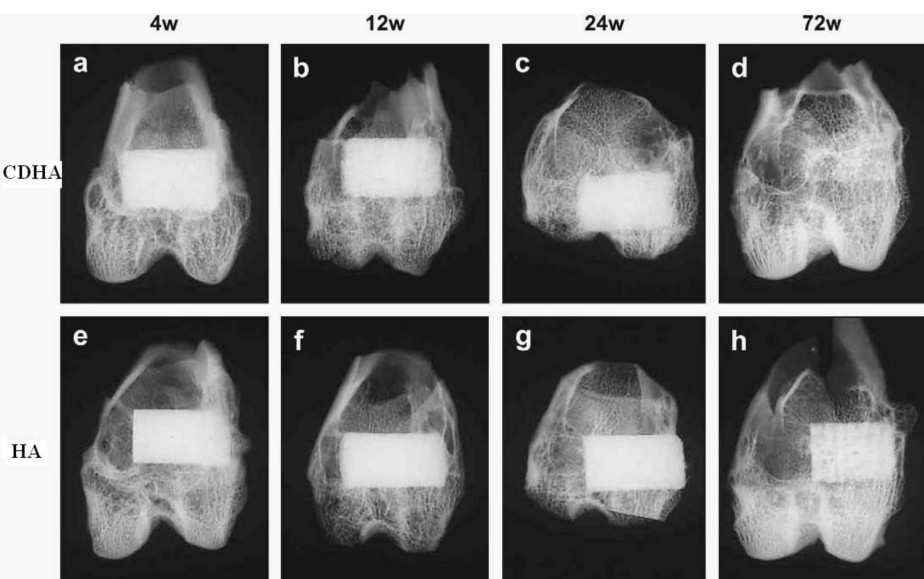

**Figure 2.** Soft X-ray photographs of the operated portion of the rabbit femur. Four weeks (**a**), 12 weeks (**b**), 24 weeks (**c**), and 72 weeks (**d**) after implantation of CDHA; 4 weeks (**e**), 12 weeks (**f**), 24 weeks (**g**), and 72 weeks (**h**) after implantation of sintered HA. Reprinted from Ref. [121] with permission.

### 3.3. Forming and Shaping

In order to fabricate $CaPO_4$ bioceramics in progressively complex shapes, scientists are investigating the use of both old and new manufacturing techniques. These techniques range from an adaptation of the age-old pottery techniques to the newest manufacturing methods for high-temperature ceramic parts for airplane engines; namely, reverse engineering [125,126] and rapid prototyping [127–129] technologies have revolutionized a generation of physical models, allowing the engineers to efficiently and accurately produce physical models and customized implants with high levels of geometric intricacy. Combined with computer-aided design and manufacturing (CAD/CAM), complex physical objects of the anatomical structure can be fabricated in a variety of shapes and sizes. In a typical application, an image of a bone defect in a patient can be taken and used to develop a three-dimensional (3D) CAD computer model [130–134]. Then, a computer can reduce the model to slices or layers. Afterwards, 3D objects and coatings are constructed layer-by-layer using rapid prototyping techniques. The examples comprise fused deposition modeling [135,136], selective laser sintering [137–142], laser cladding [143–146], 3D printing and/or plotting [73,147–153], robocasting [154–156], solid freeform fabrication [157–162], stereolithography [163–166], and direct light processing [167]. More advanced techniques, such as 4D [168,169] and 5D [170] printing techniques, have been introduced as well. Three-dimensional printing of the $CaPO_4$-based self-setting formulations is known as well [151]. Additional details of these techniques are available in the literature [171–174].

In the specific case of ceramic scaffolds, a sintering step is usually applied after printing the green bodies (see *Section 3.4. Sintering and Firing* below). Furthermore, a thermal printing process of melted $CaPO_4$ was proposed [175], while, in some cases, laser processing might be applied as well [176,177]. A schematic of the 3D-printing technique as well as some 3D-printed items are shown in Figure 3 [56]. A custom-made implant of actual dimensions would reduce the time it takes to perform the medical implantation procedure and subsequently lower the risk to the patient. Another advantage of a prefabricated, exact-fitting implant is that it can be used more effectively and applied directly to the damaged site rather than a replacement, which is formulated during surgery from a paste or granular material [158,177–179].

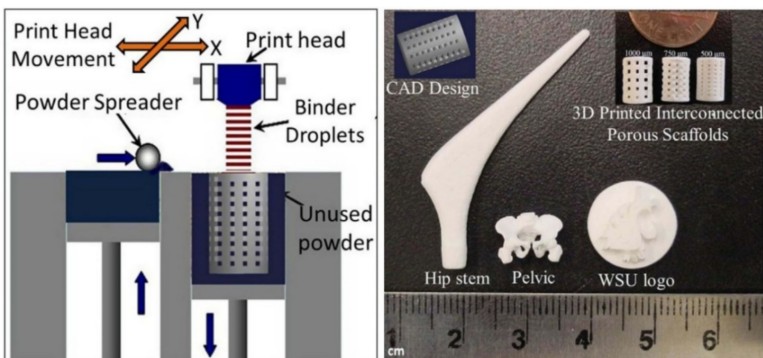

**Figure 3.** A schematic of 3D printing and some 3D-printed parts (fabricated at Washington State University) showing the versatility of 3D-printing technology for ceramic scaffolds fabrication with complex architectural features. Reprinted from Ref. [56] with permission.

In addition to the aforementioned modern techniques, classical forming and shaping approaches are still widely used. The selection of the desired technique depends greatly on the ultimate application of the bioceramic device, e.g., whether it is for a hard-tissue replacement or an integration of the device within the surrounding tissues. In general, three types of processing technologies might be used: (1) employment of a lubricant and a liquid binder with ceramic powders for shaping and subsequent firing; (2) application of self-setting and self-hardening properties of water-wet molded powders; (3) materials are melted to form a liquid and are shaped during cooling and solidification [180–182]. Since $CaPO_4$ are either thermally unstable (MCPM, MCPA, DCPA, DCPD, OCP, ACP, CDHA) or have a melting point at temperatures exceeding ~1400 °C with a partial decomposition (α-TCP, β-TCP, HA, FA, TTCP), only the first and the second consolidation approaches are used to prepare bulk bioceramics and scaffolds. The methods include uniaxial compaction [154,183,184], isostatic pressing (cold or hot) [87,185–191], granulation [192–198], loose packing [199], slip casting [75,200–205], gel casting [163,206–211], pressure mold forming [212–214], injection molding [215–218], polymer replication [219–226], ultrasonic machining [227], extrusion [228–234], and slurry dipping and spraying [235]. In addition, to form ceramic sheets from slurries, tape casting [207,236–240], doctor blade [241], and colander methods can be employed [180–182]. In addition, flexible, ultrathin (of 1 to several microns thick), freestanding HA sheets were produced by a pulsed laser deposition technique, followed by thin film isolation technology [242]. Various combinations of several techniques are also possible [77,207,243–245]. Furthermore, some of those processes might be performed under the electromagnetic field, which helps crystal aligning [201,204,246–249]. Finally, the prepared $CaPO_4$ bioceramics might be subjected to additional treatments (e.g., chemical, thermal, and/or hydrothermal ones) to convert one type of $CaPO_4$ into another one [226].

To prepare bulk bioceramics, powders are usually pressed damp in metal dies or dry in lubricated dies at pressures high enough to form sufficiently strong structures to hold together until they are sintered [250]. An organic binder, such as polyvinyl alcohol, helps to bind the powder particles altogether. Afterwards, the binder is removed by heating in air to oxidize the organic phases to carbon dioxide and water. Since many binders contain water, drying at ~100 °C is a critical step in preparing damp-formed pieces for firing. Too much or too little water in the compacts can lead to blowing apart the ware on heating or crumbling, respectively [180–182,186]. Furthermore, removal of water during drying often results in subsequent shrinkage of the product. In addition, due to local variations in water content, warping and even cracks may be developed during drying. Dry pressing and hydrostatic molding can minimize these problems [182]. Finally, the manufactured green samples are sintered.

It is important to note that forming and shaping of any ceramic products require a proper selection of the raw materials in terms of particle sizes and size distribution; namely, tough and strong bioceramics consist of pure, fine, and homogeneous microstructures. To attain this, pure powders with small average size and high surface area must be used as the

starting sources. However, for maximum packing and least shrinkage after firing, mixing of ~70% coarse and ~30% fine powders have been suggested [182]. Mixing is usually carried out in a ball mill for uniformity of properties and reaction during subsequent firing. Mechanical die forming or sometimes extrusion through a die orifice can be used to produce a fixed cross-section.

Finally, to produce the accurate shaping, necessary for the fine design of bioceramics, machine finishing might be essential [132,180,251,252]. Unfortunately, cutting tools developed for metals are usually useless for bioceramics due to their fragility; therefore, grinding and polishing appear to be the most convenient finishing techniques [132,180]. In addition, the surface of $CaPO_4$ bioceramics might be modified by various supplementary treatments [253,254], and $CaPO_4$ bioceramics might be subjected to post-processing actions, such as immersing into special solutions [255].

### 3.4. Sintering and Firing

After being formed and shaped, the $CaPO_4$ bioceramics are commonly sintered. A sintering (or firing) procedure is a thermal process in which loosely bound particles are converted into a consistent solid mass under the influence of heat and/or pressure without melting the particles. This process is of great importance to manufacture bulk bioceramics with the required mechanical properties. Usually, this technique is carried out according to controlled temperature programs of electric furnaces in adjusted ambience of air with necessary additional gasses; however, always at temperatures below the melting points of the materials. The firing step can include temporary holds at intermediate temperatures to burn out organic binders [180–182]. The heating rate, sintering temperature, and holding time depend on the starting materials. For example, in the case of HA, these values are in the ranges of 0.5–3 °C/min, 1000–1250 °C, and 2–5 h, respectively [256]. In the majority of cases, sintering allows a structure to retain its shape. However, this process might be accompanied by a considerable degree of shrinkage [257–259], which must be accommodated in the fabrication process. For instance, in the case of FA sintering, a linear shrinkage was found to occur at ~715 °C and the material reached its final density at ~890 °C. Above this value, grain growth became important and induced an intra-granular porosity, which was responsible for density decrease. At ~1180 °C, a liquid phase was formed due to formation of a binary eutectic between FA and fluorite contained in the powder as impurity. This liquid phase further promoted the coarsening process and induced formation of large pores at high temperatures [260].

In general, sintering occurs only when the driving force is sufficiently high, while the latter relates to the decrease in surface and interfacial energies of the system by matter (molecules, atoms, or ions) transport, which can proceed by solid, liquid, or gaseous phase diffusion. Namely, when solids are heated to high temperatures, their constituents are driven to move to fill up pores and open channels between the grains of powders, as well as to compensate for the surface energy differences among their convex and concave surfaces (matter moves from convex to concave). At the initial stages, bottlenecks are formed and grow among the particles (Figure 4). Existing vacancies tend to flow away from the surfaces of sharply curved necks; this is an equivalent of a material flow towards the necks, which grow as the voids shrink. Small contact areas among the particles expand and, at the same time, a density of the compact increases and the total void volume decreases. As the pores and open channels are closed during a heat treatment, the particles become tightly bonded together, and density, strength, and fatigue resistance of the sintered object improve greatly. Grain boundary diffusion was identified as the dominant mechanism for densification [261]. Furthermore, strong chemical bonds are formed among the particles, and loosely compacted green bodies are hardened to denser materials [180–182]. Further knowledge on the ceramic sintering process can be found elsewhere [262].

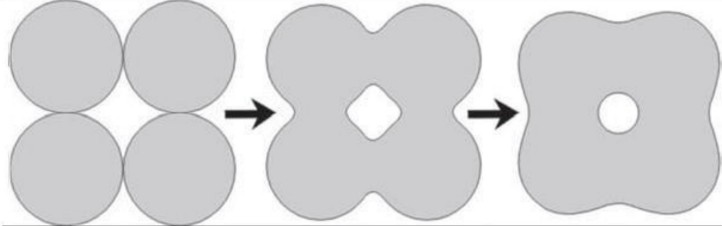

**Figure 4.** A schematic diagram representing the changes occurring with spherical particles under sintering. Shrinkage is noticeable.

In the case of $CaPO_4$, the earliest paper on their sintering was published in 1971 [263]. Since then, numerous papers on this subject have been published, and several specific processes have been found to occur during $CaPO_4$ sintering. Firstly, moisture, carbonates and all other volatile chemicals remaining from the synthesis stage, such as ammonia, nitrates, and any organic compounds, are removed as gaseous products. Secondly, unless powders are sintered, the removal of these gases facilitates production of denser ceramics with subsequent shrinkage of the samples (Figure 5). Thirdly, all chemical changes are accompanied by a concurrent increase in crystal size and a decrease in the specific surface area. Fourthly, a chemical decomposition of all acidic orthophosphates and their transformation into other phosphates (e.g., $2HPO_4^{2-} \rightarrow P_2O_7^{4-} + H_2O$) takes place. In addition, sintering causes toughening [66], densification [67,264], partial dehydroxylation (in the case of HA) [67], a partial evaporation and condensation of phosphates [265], and grain growth [261,266], as well as a mechanical strength increasing [267–269]. The latter events are due to presence of air and other gases filling gaps among the particles of unsintered powders. At sintering, the gases move towards the outside of powders, and green bodies shrink owing to decrease of distances among the particles. For example, sintering of biologically formed apatites was investigated [270,271] and the obtained products were characterized [272,273]. In all cases, the numerical value of the Ca/P ratio in sintered apatites of biological origin was higher than that of the stoichiometric HA. One should mention that in the vast majority of cases, $CaPO_4$ with Ca/P ratio < 1.5 (Table 1) are not sintered, since these compounds are thermally unstable, while sintering of nonstoichiometric $CaPO_4$ (CDHA and ACP) always leads to their transformation into various types of biphasic, triphasic, and multiphase formulations [79].

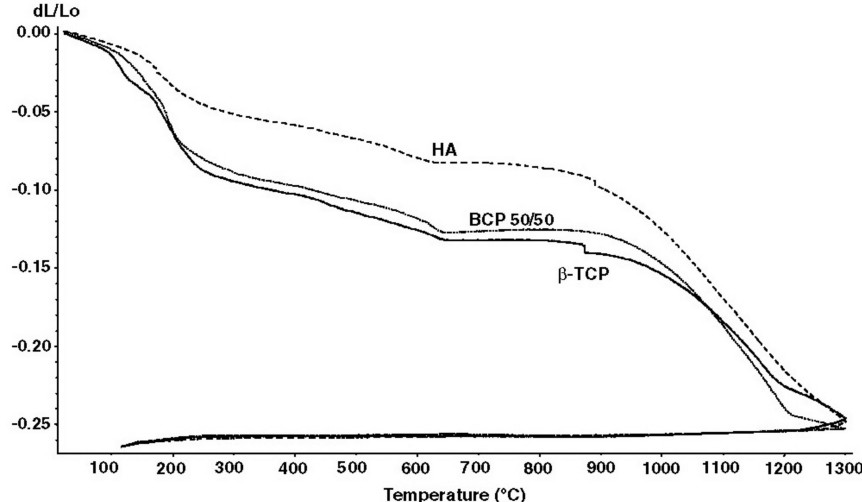

**Figure 5.** Linear shrinkage of the compacted ACP powders that were converted into β-TCP, BCP (50% HA + 50% β-TCP), and HA upon heating. According to the authors: "At 1300 °C, the shrinkage reached a maximum of approximately ~25, ~30 and ~35% for the compacted ACP powders that converted into HA, BCP 50/50 and β-TCP, respectively" [258]. Reprinted from Ref. [258] with permission.

An extensive study on the effects of sintering temperature and time on the properties of HA bioceramics revealed a correlation between these parameters and density, porosity, grain size, chemical composition, and strength of the scaffolds [274]. Namely, sintering below ~1000 °C was found to result in initial particle coalescence, with little or no densification and a significant loss of the surface area and porosity. The degree of densification appeared to depend on the sintering temperature, whereas the degree of ionic diffusion was governed by the period of sintering [274]. To enhance sinterability of $CaPO_4$, a variety of sintering additives might be added [275–278].

Solid-state pressureless sintering is the simplest procedure. For example, HA bioceramics can be pressurelessly sintered up to the theoretical density at 1000–1200 °C. Processing at even higher temperatures usually lead to exaggerated grain growth and decomposition because HA becomes unstable at temperatures exceeding ~1300 °C [6,113–117,279–281]. The decomposition temperature of HA bioceramics is a function of the partial pressure of water vapor. Moreover, processing under vacuum leads to an earlier decomposition of HA, while processing under high partial pressure of water prevents the decomposition. On the other hand, the presence of water in the sintering atmosphere was reported to inhibit densification of HA and accelerate grain growth [282]. Unexpectedly, an application of a magnetic field during sintering was found to influence the growth of HA grains [266]. A definite correlation between hardness, density, and a grain size in sintered HA bioceramics was found; despite exhibiting high bulk density, hardness started to decrease at a certain critical grain size limit [283–285].

Since grain growth occurs mainly during the final stage of sintering, to avoid this, a new method called "two-step sintering" (TSS) was proposed [286]. The method consists of suppressing grain boundary migration responsible for grain growth, while keeping grain boundary diffusion that promotes densification. The TSS approach was successfully applied to $CaPO_4$ bioceramics [77,86,287–290]. For example, HA compacts prepared from nanodimensional powders were two-step sintered. The average grain size of near full dense (>98%) HA bioceramics made via conventional sintering was found to be ~1.7 µm, while that for TSS HA bioceramics was ~190 nm (i.e., ~9 times less) with simultaneous increasing of the fracture toughness of samples from $0.98 \pm 0.12$ to $1.92 \pm 0.20$ MPa m$^{1/2}$. In addition, due to the lower second-step sintering temperature, no HA phase decomposition was detected in the TSS method [287].

Hot pressing [285,291–297], hot isostatic pressing [87,185,190,191], or hot pressing with post-sintering [298,299], as well as "cold sintering" (which is very similar to hot pressing) [300] processes make it possible to decrease the temperature of the densification process, diminish the grain size, and achieve higher densities. This leads to finer microstructures, higher thermal stability, and subsequently better mechanical properties of $CaPO_4$ bioceramics. In addition, microwave [301–306], spark plasma [69,104,307–315], flash [316,317], and ultrafast high-temperature [318] sintering techniques are alternative methods to the conventional sintering, hot pressing, and hot isostatic pressing. Both alternative methods were found to be time- and energy-efficient densification techniques. Further developments are still possible. For example, a hydrothermal hot pressing method was developed to fabricate OCP [105], CDHA [319], HA/β-TCP [294], and HA [295–298,320] bioceramics with neither thermal dehydration nor thermal decomposition. Further details on the sintering and firing processes of $CaPO_4$ bioceramics are available in the literature [115,321,322].

To conclude this section, one should note that the sintering stage is not always necessary. For example, $CaPO_4$-based bulk bioceramics with the reasonable mechanical properties might be prepared by means of self-setting (self-hardening) formulations (see *Section 6.1. Self-setting (Self-hardening) Formulations* below). Furthermore, the reader's attention is directed to an excellent review on various ceramic manufacturing techniques [323], in which various ceramic processing techniques are well described.

## 4. The Major Properties

### 4.1. Mechanical Properties

The modern generation of biomedical materials should stimulate the body's own self-repairing abilities [324]. Therefore, during healing, a mature bone should replace the modern grafts and this process must occur without transient loss of the mechanical support. Unluckily for material scientists, a human body provides one of the most inhospitable environments for the implanted biomaterials. It is warm, wet, and both chemically and biologically active. For example, a diversity of body fluids in various tissues might have a solution pH varying from 1 to 9. In addition, a body is capable of generating quite massive force concentrations, and the variance in such characteristics among individuals might be enormous. Typically, bones are subjected to ~4 MPa loads, whereas tendons and ligaments experience peak stresses in the range of 40–80 MPa. The hip joints are subjected to an average load of up to three times the body weight (3000 N), and peak loads experienced during jumping can be as high as 10 times the body weight. These stresses are repetitive and fluctuating depending on the nature of the activities, which can include standing, sitting, jogging, stretching, and climbing. Therefore, all types of implants must sustain attacks of a great variety of aggressive conditions [325]. Regrettably, there is presently no artificial material fulfilling all these requirements.

Now it is important to mention that the mechanical behavior of any ceramics is rather specific; namely, ceramics is brittle, which is attributed to high-strength ionic bonds. Thus, it is not possible for plastic deformation to happen prior to failure, as a slip cannot occur. Therefore, ceramics fail in a dramatic manner. Namely, if a crack is initiated, its progress will not be hindered by the deformation of material ahead of the crack, as would be the case in a ductile material (e.g., a metal). In ceramics, the crack will continue to propagate, rapidly resulting in a catastrophic breakdown. In addition, the mechanical data typically have a considerable amount of scatter [181]. Alas, all of these are applicable to $CaPO_4$ bioceramics.

For dense bioceramics, the strength is a function of the grain sizes. Namely, finer-grain-size bioceramics have smaller flaws at the grain boundaries and thus are stronger than ones with larger grain sizes. Thus, in general, the strength for ceramics is proportional to the inverse square root of the grain sizes [326]. In addition, the mechanical properties decrease significantly with increasing content of an amorphous phase, microporosity, and grain sizes, while a high crystallinity, a low porosity, and small grain sizes tend to give a higher stiffness, a higher compressive and tensile strength, and a greater fracture toughness. Furthermore, ceramics strength appears to be very sensitive to slow crack growth [327]. Accordingly, from the mechanical point of view, $CaPO_4$ bioceramics appear to be brittle polycrystalline materials for which the mechanical properties are governed by crystallinity, grain size, grain boundaries, porosity, and composition [328]. Thus, it possesses poor mechanical properties (for instance, a low impact and fracture resistances) that do not allow $CaPO_4$ bioceramics to be used in load-bearing areas, such as artificial teeth or bones [50–53]. For example, fracture toughness (this is a property that describes the ability of a material containing a crack to resist fracture and is one of the most important properties of any material for virtually all design applications) of HA bioceramics does not exceed the value of ~1.2 MPa·m$^{1/2}$ [329] (human bone: 2–12 MPa·m$^{1/2}$). It decreases exponentially with increasing porosity [330]. Generally, fracture toughness increases with grain size decreasing. However, in some materials, especially noncubic ceramics, fracture toughness reaches the maximum and rapidly drops with decreasing grain size. For example, a fracture toughness of pure hot-pressed HA with grain sizes between 0.2–1.2 μm was investigated. The authors found two distinct trends, where fracture toughness decreased with increasing grain size above ~0.4 μm and subsequently decreased with decreasing grain size. The maximum fracture toughness measured was 1.20 ± 0.05 MPa·m$^{1/2}$ at ~0.4 μm [291]. Fracture energy of HA bioceramics is in the range of 2.3–20 J/m$^2$, while the Weibull modulus (a measure of the spread or scatter in fracture strength) is low (~5–12) in wet environments, which means that HA behaves as a typical brittle ceramics and indicates a low reliability of HA implants [331]. Porosity has a great influence on the Weibull modulus [332,333]. In addition,

the reliability of HA bioceramics was found to depend on deformation mode (bending or compression), along with pore size and pore size distribution: reliability was higher for smaller average pore sizes in bending but lower for smaller pore sizes in compression [334]. Interestingly, three peaks of internal friction were found at temperatures of about –40, 80, and 130 °C for HA but no internal friction peaks were obtained for FA in the measured temperature range; this effect was attributed to the differences of $F^-$ and $OH^-$ positions in FA and HA, respectively [335]. Differences in internal friction values were also found between HA and TCP [336].

Bending, compressive, and tensile strengths of dense HA bioceramics are in the ranges of 38–250, 120–900, and 38–300 MPa, respectively. Similar values for porous HA bioceramics are substantially lower: 2–11, 2–100, and ~3 MPa, respectively [331]. These wide variations in the properties are due to both structural variations (e.g., an influence of remaining microporosity, grain sizes, presence of impurities, etc.) and manufacturing processes, and they are also caused by a statistical nature of the strength distribution. Strength was found to increase with Ca/P ratio increasing, reaching the maximum value around Ca/P ~1.67 (stoichiometric HA) and decreasing suddenly when Ca/P > 1.67 [331]. Furthermore, strength decreases almost exponentially with increasing porosity [337,338]. However, by changing the pore geometry, it is possible to influence the strength of porous bioceramics. It is also worth mentioning that porous $CaPO_4$ bioceramics are considerably less fatigue-resistant than dense ones (in materials science, fatigue is the progressive and localized structural damage that occurs when a material is subjected to cyclic loading). Both grain sizes and porosity are reported to influence the fracture path, which itself has little effect on the fracture toughness of $CaPO_4$ bioceramics [328,339]. However, no obvious decrease in mechanical properties was found after $CaPO_4$ bioceramics had been aged in the various solutions during the different periods of time [340].

Young's (or elastic) modulus of dense HA bioceramics is in the range of 3–120 GPa [341,342], which is more or less similar to those of the most resistant components of the natural calcified tissues (dental enamel: ~74 GPa, dentine: ~21 GPa, compact bone: ~18–22 GPa). This value depends on porosity [343,344]. Nevertheless, dense bulk compacts of HA have mechanical resistances of the order of 100 MPa versus ~00 MPa of human bones, drastically diminishing their resistances in the case of porous bulk compacts [345]. Young's modulus measured in bending is between 44 and 88 GPa. To investigate the subject in more detail, various types of modeling and calculations are increasingly used [346–350]. For example, the elastic properties of HA appeared to be significantly affected by the presence of vacancies, which softened HA via reducing its elastic modules [350]. In addition, a considerable anisotropy in the stress–strain behavior of the perfect HA crystals was found by ab initio calculations [347]. The crystals appeared to be brittle for tension along the *z*-axis with the maximum stress of ~9.6 GPa at 10% strain. Furthermore, the structural analysis of the HA crystal under various stages of tensile strain revealed that the deformation behavior manifested itself mainly in the rotation of $PO_4$ tetrahedrons with concomitant movements of both the columnar and axial Ca ions [347]. Data for single crystals are also available [351]. Vickers hardness (a measure of the resistance to permanent indentation) of dense HA bioceramics is within 3–7 GPa, while the Poisson's ratio (the ratio of the contraction or transverse strain to the extension or axial strain) for HA is about 0.27, which is close to that of bones (~0.3). At temperatures within 1000–1100 °C, dense HA bioceramics were found to exhibit superplasticity with a deformation mechanism based on grain boundary sliding [312,352,353]. Furthermore, both wear resistance and friction coefficient of dense HA bioceramics are comparable to those of dental enamel [331].

Due to a high brittleness (associated with a low crack resistance), the biomedical applications of $CaPO_4$ bioceramics are focused on production of non-load-bearing implants, such as pieces for middle ear surgery, filling of bone defects in oral or orthopedic surgery, and coating of dental implants and metallic prosthesis (see below) [62,354,355]. Therefore, methods are continuously sought to improve the reliability of $CaPO_4$ bioceramics. Namely, the mechanical properties of sintered bioceramics might be improved by changing the

morphology of the initial $CaPO_4$ [356]. In addition, diverse reinforcements (ceramics, metals, or polymers) have been applied to manufacture various biocomposites and hybrid biomaterials [357], but that is another story. However, successful hybrid formulations consisting of $CaPO_4$ only [358–365] are within the scope of this review. Namely, bulk HA bioceramics might be reinforced by HA whiskers [359–363]. Furthermore, various biphasic apatite/TCP formulations were tested [358,364,365] and, for example, a superior superplasticity of HA/β-TCP biocomposites to HA bioceramics was detected [364].

Another method to improve the mechanical properties of $CaPO_4$ bioceramics is to cover the items by polymeric coatings [366–368] or infiltrate porous structures by polymers [369–371]; however, this is another topic. Other approaches are also possible [154]. Further details on the mechanical properties of $CaPO_4$ bioceramics are available elsewhere [330,331,372], where interested readers are referred.

### 4.2. Electric/Dielectric and Piezoelectric Properties

Recently, an interest in both electric/dielectric [301,373–385] and piezoelectric [386,387] properties of $CaPO_4$ bioceramics has been expressed. In addition, some types of $CaPO_4$ bioceramics (namely, HA) appear to be electrets [388,389]. An electret is a dielectric material that has a quasi-permanent electric charge or dipole polarization. An electret generates internal and external electric fields, and is the electrostatic equivalent of a permanent magnet [390]. For example, a surface ionic conductivity of both porous and dense HA bioceramics was examined for humidity sensor applications, since the room temperature conductivity was influenced by relative humidity [374]. Namely, the ionic conductivity of HA is a subject of research for its possible use as a gas sensor for alcohol [375], carbon dioxide [373,382], or carbon monoxide [378]. Electric measurements were also used as a characterization tool to study the evolution of microstructure in HA bioceramics [376]. More to the point, the dielectric properties of HA were examined to understand its decomposition to β-TCP [375]. In the case of CDHA, the electric properties, in terms of ionic conductivity, were found to increase after compression of the samples at 15 t/cm$^2$, which was attributed to establishment of some order within the apatitic network [377]. The conductivity mechanism of CDHA appeared to be multiple [380]. Furthermore, there are attempts to develop HA and/or CDHA electrets for biomedical utilization [379,388,389].

The electric properties of $CaPO_4$ bioceramics appear to influence their biomedical applications. For example, there is an interest in polarization of HA bioceramics to generate a surface charge by the applying a constant DC electric field of 0.5–10.0 kV/cm at elevated temperatures (300–1000 °C) to samples previously sintered at ~ 1000–1250 °C for ~2 h. This technique is called thermally stimulated polarization and its results indicated that the polarization effects were a consequence of electrical dipoles associated with the formation of defects inside crystal grains, such as thermally-induced $OH^-$ vacancies, and of the space charge polarization that originated in the grain boundaries [391–393]. The presence of surface charges on HA was shown to have a significant effect on both in vitro and in vivo crystallization of biological apatite [394–400], as well as on an ability to adsorb various types of phosphate ions [393]. Furthermore, a growth of both biomimetic $CaPO_4$ and bones was found to be accelerated on negatively charged surfaces and decelerated at positively charged surfaces [398–407]. A similar effect was found for adsorption of bovine serum albumin [408]. In addition, the electric polarization of $CaPO_4$ was found to accelerate a cytoskeleton reorganization of osteoblast-like cells [409–412], extend bioactivity [413], enhance bone ingrowth through the pores of porous implants [414], and influence the cell activity [415,416]. The positive effect of electric polarization was found for carbonated apatite as well [417]. There is an interesting study on the interaction of a blood coagulation factor on electrically polarized HA surfaces [418]. Further details on the electric properties of $CaPO_4$-based bioceramics are available in the literature [301,383,384,389].

### 4.3. Possible Transparency

Single crystals of all types of $CaPO_4$ are optically transparent for the visible light. As bioceramics of $CaPO_4$ have a polycrystalline nature with a random orientation of big amounts of small crystals, it is opaque and of white color, unless colored dopants have been added. However, in some cases, a transparency is convenient to provide some essential advantages (e.g., to enable direct viewing of living cells, their attachment, spreading, proliferation, and osteogenic differentiation cascade in a transmitted light). Thus, transparent $CaPO_4$ bioceramics (Figure 6) [419] have been prepared and investigated [69,87,185,187,310–315,419–427]. They can exhibit an optical transmittance of ~66% at a wavelength of 645 nm [425]. The preparation techniques include a hot isostatic pressing [87,185,187,426], an ambient-pressure sintering [420], a gel casting coupled with a low-temperature sintering [421,424], and a pulse electric current sintering [422], as well as spark plasma [69,307–315] and flash [316,317] sintering techniques. Fully dense, transparent $CaPO_4$ bioceramics are obtained at temperatures above ~800 °C. Depending on the preparation technique, the transparent bioceramics have a uniform grain size ranging from ~67 nm [87] to ~250 μm [421] and are always pore-free. Furthermore, translucent $CaPO_4$ bioceramics are also known [87,263,428–430]. Concerning possible biomedical applications, the optically transparent in visible light $CaPO_4$ bioceramics can be useful for direct viewing of other objects, such as cells, in some specific experiments [423]. In addition, the transparency for laser light $CaPO_4$ bioceramics may appear to be convenient for minimal invasive surgery by allowing passing the laser beam through it to treat the injured tissues located underneath. However, due to a lack of both porosity and the necessity to have see-through implants inside the body, the transparent and translucent forms of $CaPO_4$ bioceramics will hardly be extensively used in medicine, except for the aforementioned cases and possible eye implants.

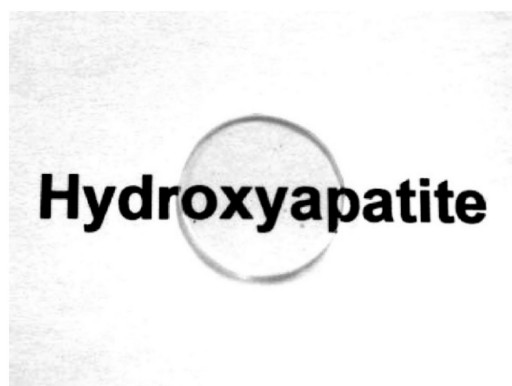

**Figure 6.** Transparent HA bioceramics prepared by spark plasma sintering at 900 °C from nanosized HA single crystals. Reprinted from Ref. [419] with permission.

### 4.4. Porosity

Porosity is defined as a percentage of voids in solids, and this morphological property is independent of the material. The surface area of porous bodies is much higher, which guarantees a good mechanical fixation in addition to providing sites on the surface that allow chemical bonding between the bioceramics and bones [431]. Furthermore, a porous material may have both closed (isolated) pores and open (interconnected) pores. The latter look similar to tunnels and are accessible by gases, liquids, and particulate suspensions [432]. The open-cell nature of porous materials (also known as reticulated materials) is a unique characteristic essential in many applications. In addition, pore dimensions are also important. Namely, the dimensions of open pores are directly related to bone formation, since such pores grant both the surface and space for cell adhesion and bone ingrowth [433–435]. On the other hand, pore interconnection provides the ways for cell distribution and migration, and it allows an efficient in vivo blood vessel formation suitable for sustaining bone tissue neo-formation and possibly remodeling [123,414,436–440]. Namely,

porous CaPO$_4$ bioceramics are colonized easily by cells and bone tissues [436,439,441–446]. Therefore, interconnecting macroporosity (pore size > 100 μm) [84,431,436,447,448] is intentionally introduced in solid bioceramics (Figure 7). Calcining of natural bones and teeth appears to be the simplest way to prepare porous CaPO$_4$ bioceramics [7–14]. In addition, macroporosity might be formed artificially due to a release of various easily removable compounds and, for that reason, incorporation of pore-creating additives (porogens) is the most popular technique to create macroporosity. The porogens are crystals, particles, or fibers of either volatile (they evolve gases at elevated temperatures) or soluble substances. The popular examples comprise paraffin [449–451], naphthalene [328,452–454], sucrose [455,456], NaHCO$_3$ [457–459], NaCl [460,461], polymethylmethacrylate [74,462–464], hydrogen peroxide [465–468], cellulose [469], and its derivatives [64]. Several other compounds [338,470–477], including carbon nanotubes [478], might be used as porogens as well. The ideal porogen should be nontoxic and be removed at ambient temperature, thereby allowing the bioceramic/porogen mixture to be injected directly into a defect site and allowing the scaffold to fit the defect [479]. Sintering particles, preferably spheres of equal size, is a similar way to generate porous 3D bioceramics of CaPO$_4$; however, pores resulting from this method are often irregular in size and shape and not fully interconnected with one another. Schematic drawings of various types of the ceramic porosity are shown in Figure 8 [480]. One should note that 3D-printing techniques allow producing structures with tailored pore orientations by changing framework directions in a controlled, periodic pattern (Figure 9) [481].

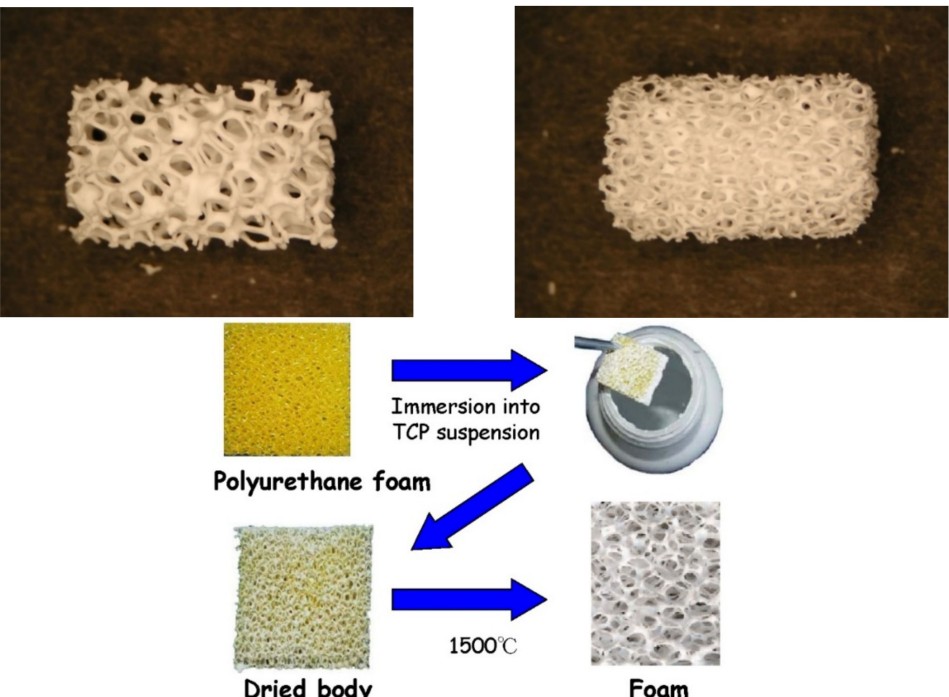

**Figure 7.** Photographs of commercially available porous CaPO$_4$ bioceramics with different porosity (**top**) and a method of their production (**bottom**). For photos, the horizontal field width is 20 mm.

Many other techniques, such as replication of polymer foams by impregnation [219–221, 224,482–486] (Figure 7), various types of casting [202,203,207,209,468,487–495], suspension foaming [101], surfactant washing [496], microemulsions [497,498], and ice templating [499–502], as well as many other approaches [68,71,74,75,140,503–528], have been applied to fabricate porous CaPO$_4$ bioceramics. Some of them are summarized in Table 2 [479]. In addition, both natural CaCO$_3$ porous materials, such as coral skeletons [529,530], shells [530,531], and even wood [532], as well as artificially prepared ones [533], can be converted into porous CaPO$_4$ under the hydrothermal conditions (250 °C, 24–48 h) with the microstructure undamaged. Porous HA bioceramics can also be obtained by

hydrothermal hot pressing. This technique allows solidification of the HA powder at 100–300 °C (30 MPa, 2 h) [320]. In another approach, bi-continuous water-filled microemulsions are used as preorganized systems for the fabrication of needle-like frameworks of crystalline HA (2 °C, 3 weeks) [497,498]. In addition, porous CaPO$_4$ might be prepared by a combination of gel casting and foam burn out methods [243,245], as well as by hardening of the self-setting formulations [450,451,458–461,520]. Lithography was used to print a polymeric material, followed by packing with HA and sintering [507]. Hot pressing was applied as well [292,293]. More to the point, an HA suspension can be cast into a porous CaCO$_3$ skeleton, which is then dissolved, leaving a porous network [503]. A 3D periodic macroporous frame of HA was fabricated via a template-assisted colloidal processing technique [509,512]. In addition, porous HA bioceramics might be prepared by using different starting HA powders and sintering at various temperatures by a pressureless sintering [505]. Porous bioceramics with an improved strength might be fabricated from CaPO$_4$ fibers or whiskers. In general, fibrous porous materials are known to exhibit an improved strength due to fiber interlocking, crack deflection, and/or pullout [534]. Namely, porous bioceramics with well-controlled open pores were processed by sintering of fibrous HA particles [504]. In another approach, porosity was achieved by firing apatite-fiber compacts mixed with carbon beads and agar. By varying the compaction pressure, firing temperature and carbon/HA ratio, the total porosity was controlled in the ranges from ~40% to ~85% [64]. Finally, a superporous (~85% porosity) HA bioceramic was developed as well [515,517,518]. Additional information on the processing routes to produce porous ceramics can be found in the literature [535].

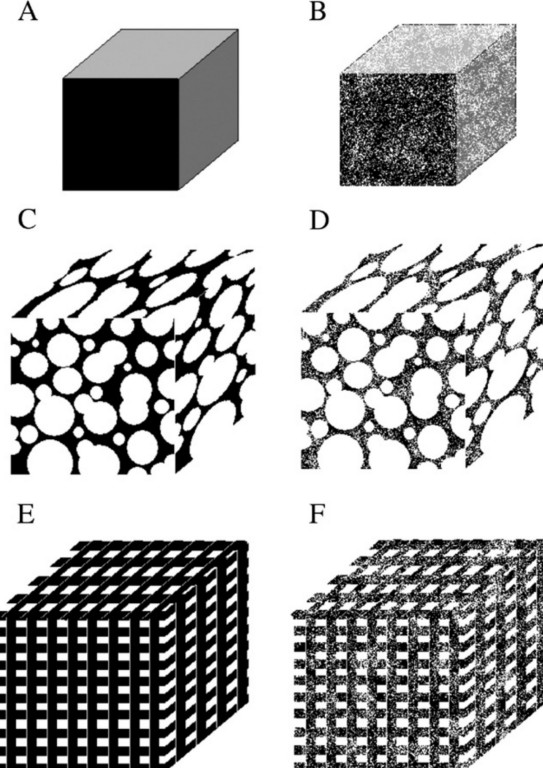

**Figure 8.** Schematic drawings of various types of the ceramic porosity: (**A**)—nonporous, (**B**)—microporous, (**C**)—macroporous (spherical), (**D**)—macroporous (spherical) + micropores, (**E**)—macroporous (3D-printing), (**F**)— macroporous (3D-printing) + micropores. Reprinted from Ref. [480] with permission.

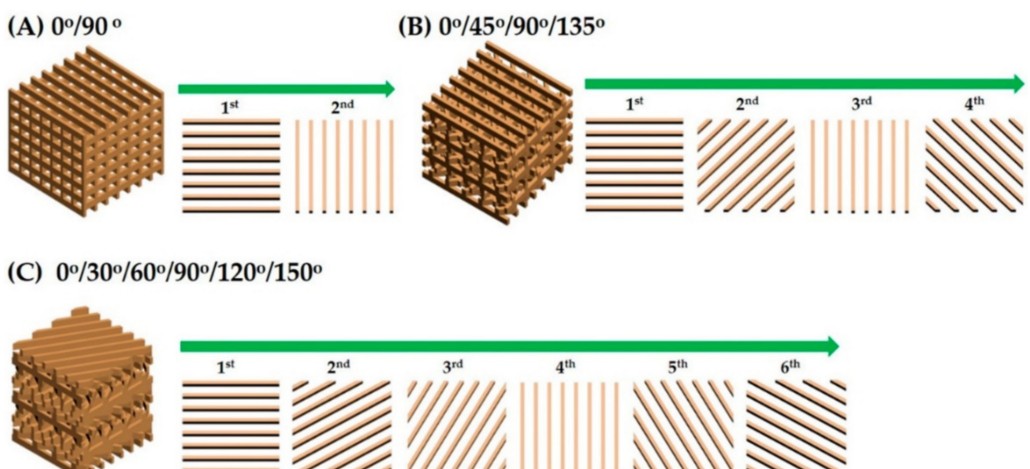

**Figure 9.** A schematic diagram showing the porous structures of various CaPO4 scaffolds with different pore orientations: (**A**) 0°/90°, (**B**) 0°/45°/90°/135°, and (**C**) 0°/30°/60°/90°/120°/150°, with their top-down view of the repeating CaPO4 frameworks. Reprinted from Ref. [481] with permission.

Bioceramic microporosity (pore size < 10 μm), which is defined by its capacity to be impregnated by biological fluids [536], results from the sintering process, while the pore dimensions mainly depend on the material composition, thermal cycle, and sintering time. The microporosity provides both a greater surface area for protein adsorption and increased ionic solubility. For example, embedded osteocytes distributed throughout microporous rods might form a mechanosensory network, which would not be possible in scaffolds without microporosity [537,538]. CaPO$_4$ bioceramics with nanodimensional (<100 nm) pores might be fabricated as well [539–543]. It is important to stress that differences in porogens usually influence the bioceramics' macroporosity, while differences in sintering temperatures and conditions affect the percentage of microporosity. Usually, the higher the sintering temperature, the lower both the microporosity content and the specific surface area of bioceramics. Namely, HA bioceramics sintered at ~1200 °C show significantly less microporosity and a dramatic change in crystal sizes, if compared with those sintered at ~1050 °C (Figure 10) [544]. Furthermore, the average shape of pores was found to transform from strongly oblate to round at higher sintering temperatures [545]. The total porosity (macroporosity + microporosity) of CaPO$_4$ bioceramics was reported to be ~70% [546] or even ~85% [515,517,518] of the entire volume. In the case of coralline HA or bovine-derived apatites, the porosity of the original biologic material (coral or bovine bone) is usually preserved during processing [547]. To finalize the production topic, creation of the desired porosity in CaPO$_4$ bioceramics is a rather complicated engineering task and interested readers are referred to the additional publications on the subject [338,435,519,548–553].

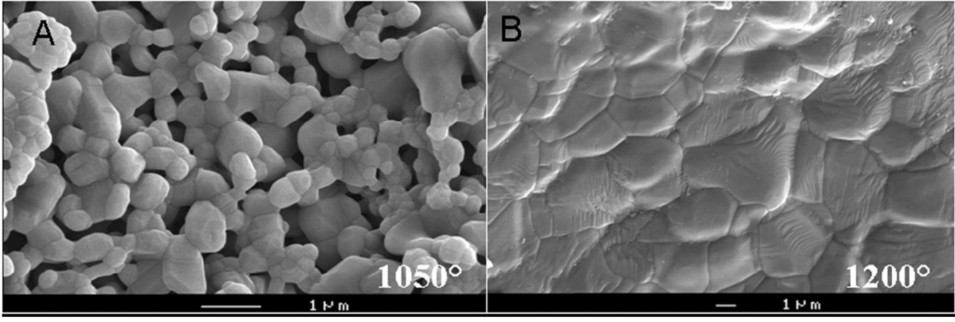

**Figure 10.** SEM pictures of HA bioceramics sintered at (**A**) 1050 °C and (**B**) 1200 °C. Note the presence of microporosity in **A** and not in **B**. Reprinted from Ref. [544] with permission.

**Table 2.** The procedures used to manufacture porous CaPO$_4$ scaffolds for tissue engineering [479].

| Year | Location | Process | Apatite from: | Sintering | Compressive Strength | Pore Size | Porosity | Method of Porosity Control |
|---|---|---|---|---|---|---|---|---|
| 2006 | Deville et al., Berkeley, CA | HA + ammonium methacrylate in polytetrafluoroethylene mold, freeze dried and sintered. | HA #30 | Yes: 1300 °C | 16 MPa 65 MPa 145 MPa | Open unidirectional 50–150 µm. | >60% 56% 47% | Porosity control: slurry conc. Structure controlled by physics of ice front formation. |
| 2006 | Saiz et al., Berkeley, CA | Polymer foams coated, compressed after infiltration, then calcined. | HA powder | Yes: 700–1300 °C | – | 100–200 µm. | – | Porosity control: extent of compression, HA loading. |
| 2006 | Murugan et al., Singapore + USA | Bovine bone cleaned, calcined. | bovine bone | Yes: 500 °C | – | Retention of nanosized pores. | – | Porosity control: native porosity of bovine bone. |
| 2006 | Xu et al., Gaithersburg, MD | Directly injectable CaPO$_4$ cement, self-hardens, mannitol as porogen. | nanocrystalline HA | No | 2.2–4.2 MPa (flexural) | 0%–50% macroporous. | 65%–82% | Porosity control: mannitol mass fraction in mixture. |
| 2004 | Landi et al., Italy + Indonesia | Sponge impregnation, isotactic pressing, sintering of HA in simulated body fluid. | CaO + H$_3$PO$_4$ | Yes: 1250 °C for 1 h | 23 ± 3.8 MPa | Closed 6%, open 60%. | 66% | Porosity control: possibly by controlling HA particle size. Not suggested by authors. |
| 2003 | Charriere et al., EPFL, Switzerland | Thermoplastic negative porosity by Inkjet printing, slip casting process for HA. | DCPA + calcite | No: 90 °C for 1 day. | 12.5 ± 4.6 MPa | – | 44% | Porosity control: negative printing. |
| 2003 | Almirall et al., Barcelona, Spain | α-TCP foamed with hydrogen peroxide at different conc., liq. Ratios, poured in polytetrafluoroethylene molds. | A-TCP + (10% and 20% H$_2$O$_2$) | No: 60 °C for 2 h. | 1.41 ± 0.27 MPa 2.69 ± 0.91 MPa | 35.7% macro. 29.7% micro. 26.8% macro. 33.8% micro. | 65.5% 60.7% | Porosity control: different concentration, α-TCP particle sizes. |
| 2003 | Ramay et al., Seattle, WA | Slurries of HA prepared: gel-casting + polymer sponge technique, sintered. | HA powder | Yes: 600 °C for 1 h 1350 °C for 2·h. | 0.5–5 MPa | 200–400 µm. | 70%–77% | Porosity control: replicate of polymer sponge template. |

**Table 2.** *Cont.*

| Year | Location | Process | Apatite from: | Sintering | Compressive Strength | Pore Size | Porosity | Method of Porosity Control |
|------|----------|---------|---------------|-----------|---------------------|-----------|----------|---------------------------|
| 2003 | Miao et al., Singapore | TTCP to $CaPO_4$ cement. Slurry cast on polymer foam, sintered. | TTCP | Yes: 1200 °C for 2 h. | – | 1 mm macro. 5 μm micro. | ~70% | Porosity control: Recoating time, polyurethane foam. |
| 2003 | Uemura et al., China + Japan | Slurry of HA with polyoxyethylene lauryl ether (cross-linked) and sintered. | HA powders | Yes: 1200 °C for 3 h. | 2.25 MPa (0 wk) 4.92 MPa (12 wk) 11.2 MPa (24 wx) | 500 μm. 200 μm interconnects. | ~77% | Porosity control: polymer interconnects cross-linking. |
| 2003 | Ma et al., Singapore + USA | Electrophoretic deposition of HA, sintering. | HA powders | Yes: 1200 °C for 2 h. | 860 MPa | 0.5 μm. 130 μm. | ~20% | Porosity control: electrophoresis field. |
| 2002 | Barralet et al., Birmingham, London, UK | $CaPO_4$ cement + sodium phosphate ice, evaporated. | $CaCO_3$ + DCPD | 1st step: 1400 °C for 1 day. | $0.6 \pm 0.27$ MPa | 2 μm. | $62\% \pm 9\%$ | Porosity control: porogen shape. |

Regarding the biomedical importance of porosity, studies revealed that increasing of both the specific surface area and pore volume of bioceramics might greatly accelerate the in vivo process of apatite deposition and, therefore, enhance the bone-forming bioactivity. More importantly, a precise control over the porosity, pore dimensions, and internal pore architecture of bioceramics on different length scales is essential for understanding the structure–bioactivity relationship and the rational design of better bone-forming biomaterials [551,554,555]. Namely, in antibiotic charging experiments, $CaPO_4$ bioceramics with nanodimensional (<100 nm) pores showed a much higher charging capacity (1621 μg/g) than those of commercially available $CaPO_4$ (100 μg/g), which did not contain nanodimensional porosity [549]. In other experiments, porous blocks of HA were found to be viable carriers with sustained release profiles for drugs [556] and antibiotics over 12 days [557] and 12 weeks [558], respectively. Unfortunately, porosity significantly decreases the strength of implants [334,339,372]. Thus, porous $CaPO_4$ implants cannot be loaded and are used to fill only small bone defects; however, their strength increases gradually when bones ingrow into the porous network of $CaPO_4$ implants [119,559–562]. For example, bending strengths of 4–60 MPa for porous HA implants filled with 50%–60% of cortical bone were reported [559], while in another study an ingrown bone increased strength of porous HA bioceramics by a factor of three to four [561].

Unfortunately, the biomedical effects of bioceramics' porosity are not straightforward. For example, the in vivo response of $CaPO_4$ to different porosity was investigated, and a hardly any effect of macropore dimensions (~150, ~260, ~510, and ~1220 μm) was observed [563]. In another study, a greater differentiation of mesenchymal stem cells was observed when cultured on ~200 μm pore size HA scaffolds when compared to those on ~500 μm pore size HA [564]. The latter finding was attributed to the fact that a higher pore volume in ~500 μm macropore scaffolds might contribute to a lack of cell confluency, leading to the cells proliferating before beginning differentiation. In addition, the authors hypothesized that bioceramics having less than the optimal pore dimensions induced quiescence in differentiated osteoblasts due to reduced cell confluency [564]. In still another study, the use of BCP (HA/TCP = 65/35 wt.%) scaffolds with cubic pores of ~500 μm resulted in the highest bone formation compared with the scaffolds with lower (~100 μm) or higher (~1000 μm) pore sizes [565]. Furthermore, $CaPO_4$ bioceramics with greater strut porosity appeared to be more osteoinductive [566]. As early as 1979, Holmes suggested that the optimal pore range was 200–400 μm with the average human osteon size of ~223 μm [567]. In 1997, Tsurga and coworkers implied that the optimal pore size of bioceramics that supported ectopic bone formation was 300–400 μm [568]. Thus, there is no need to create $CaPO_4$ bioceramics with very big pores; however, the pores must be interconnected [437,447,448,569]. Interconnectivity governs a depth of cells or tissue penetration into the porous bioceramics, and it allows development of blood vessels required for new bone nourishing and wastes removal [570,571]. Nevertheless, the total porosity of implanted bioceramics appears to be important. For example, 60% porous β-TCP granules achieved a higher bone fusion rate than 75% porous β-TCP granules in lumbar posterolateral fusion [537].

More details on the importance of $CaPO_4$ bioceramics porosity on bone regeneration are available in a topical review [572].

## 5. Biological Properties and In Vivo Behavior

The most important differences between bioactive bioceramics and all other implanted materials comprise inclusion in the metabolic processes of the organism, adaptation of either surface or the entire material to the biomedium, integration of a bioactive implant with bone tissues at the molecular level, or the complete replacement of a resorbable bioceramics by healthy bone tissues. All of the enumerated processes are related to the effect of an organism on the implant. Nevertheless, another aspect of implantation is also important—the effect of the implant on the organism. For example, use of bone implants from corpses or animals, even after they have been treated in various ways, provokes a

substantially negative immune reaction in the organism, which substantially limits the application of such implants. In this connection, it is useful to dwell on the biological properties of bioceramic implants, particularly those of $CaPO_4$, which in the course of time may be resorbed completely [573].

*5.1. Interactions with Surrounding Tissues and the Host Responses*

All interactions between implants and the surrounding tissues are dynamic processes. Water, dissolved ions, various biomolecules, and cells surround the implant surface within the initial few seconds after the implantation. It is accepted that no foreign material placed inside a living body is completely compatible. The only substances that conform completely are those manufactured by the body itself (autogenous), while any other substance, which is recognized as foreign, initiates some types of reactions (a host-tissue response). The reactions occurring at the biomaterial/tissue interfaces lead to time-dependent changes in the surface characteristics of both the implanted biomaterials and the surrounding tissues [58,574].

In order to develop new biomaterials, it is necessary to understand the in vivo host responses. Similar to any other species, biomaterials and bioceramics react chemically with their environment and, ideally, they should neither induce any changes nor provoke undesired reactions in the neighboring or distant tissues. In general, living organisms can treat artificial implants as biotoxic (or bioincompatible [53]), bioinert (or biostable [47]), biotolerant (or biocompatible [53]; however, this term appears to be questionable [575]), and bioactive and bioresorbable materials [1–3,42,43,50–53,573,574,576]. Biotoxic (e.g., alloys containing cadmium, vanadium, lead, and other toxic elements) materials release to the body substances in toxic concentrations and/or trigger the formation of antigens that may cause immune reactions ranging from simple allergies to inflammation to septic rejection with the associated severe health consequences. They cause atrophy, pathological change, or rejection of living tissue near the material as a result of chemical, galvanic, or other processes. Bioinert (this term should be used with care, since it is clear that any material introduced into the physiological environment will induce a response; however, for the purposes of biomedical implants, the term can be defined as a minimal level of response from the host tissue), such as zirconia, alumina, carbon, and titanium, as well as biotolerant (e.g., polymethylmethacrylate, titanium, and Co–Cr alloy), materials do not release any toxic constituents but also do not show positive interaction with living tissue. They evoke a physiological response to form a fibrous capsule, thus isolating the material from the body. In such cases, thickness of the layer of fibrous tissue separating the material from other tissues of an organism can serve as a measure of bioinertness. Generally, both bioactivity and bioresorbability phenomena are fine examples of chemical reactivity, and $CaPO_4$ (both nonsubstituted and ion-substituted ones) fall into these two categories of bioceramics [1–3,42,43,50–53,573,574,576]. A bioactive material will dissolve slightly but promote formation of a surface layer of biological apatite before interfacing directly with the tissue at the atomic level, that results in formation of direct chemical bonds to bones. Such implants provide a good stabilization for materials that are subject to mechanical loading. A bioresorbable material will dissolve over time (regardless of the mechanism leading to the material removal) and allow a newly formed tissue to grow into any surface irregularities, but may not necessarily interface directly with the material. Consequently, the functions of bioresorbable materials are to participate in dynamic processes of formation and reabsorption occurring in bone tissues; thus, bioresorbable materials are used as scaffolds or filling spacers, allowing the tissues their infiltration and substitution [180,576–579].

It is important to stress that a distinction between the bioactive and bioresorbable bioceramics might be associated with structural factors only. Namely, bioceramics made from nonporous, dense, and highly crystalline HA behave as bioinert (but a bioactive) materials and are retained in an organism for at least 5–7 years without noticeable changes (Figure 2 bottom), while highly porous bioceramics of the same composition can be resorbed approximately within a year. Furthermore, submicron-sized HA powders are biodegraded

even faster than the highly porous HA scaffolds. Other examples of bioresorbable materials comprise porous bioceramic scaffolds made of biphasic, triphasic, or multiphasic CaPO$_4$ formulations [79] or bone grafts (dense or porous) made of CDHA [121], TCP [74,580,581], and/or ACP [470,582]. One must note that at the beginning of the 2000s, the concepts of bioactive and bioresorbable materials were converged and bioactive materials were made bioresorbable, while bioresorbable ones were made bioactive [583].

Although in certain in vivo experiments inflammatory reactions were observed after implantation or injection of CaPO$_4$ [584–593], the general conclusion on using CaPO$_4$ with Ca/P ionic ratio within 1.0–1.7 is that all types of implants (bioceramics of various porosities and structures, powders, or granules) are not only nontoxic but also induce neither inflammatory nor foreign-body reactions [108,571,594]. The biological response to implanted CaPO$_4$ follows a similar cascade to that observed in fracture healing. This cascade includes a hematoma formation, inflammation, neovascularization, osteoclastic resorption, and a new bone formation. An intermediate layer of fibrous tissue between the implants and bones has been never detected. Furthermore, CaPO$_4$ implants display the ability to directly bond to bones [1–3,42,43,50–53,573,574,576]. For further details, interested readers are referred to a good review on cellular perspectives of bioceramic scaffolds for bone tissue engineering [479].

One should note that the aforementioned rare cases of the inflammatory reactions to CaPO$_4$ bioceramics were often caused by "other" reasons. For example, a high rate of wound inflammation occurred when highly porous HA was used. In that particular case, the inflammation was explained by sharp implant edges, which irritated surrounding soft tissues [585]. To avoid this, only rounded material should be used for implantation (Figure 11) [595]. Another reason for inflammation produced by porous HA could be due to micro movements of the implants, leading to simultaneous disruption of a large number of microvessels, which grow into the pores of the bioceramics. This would immediately produce an inflammatory reaction. Additionally, problems could arise in clinical tests connected with migration of granules used for alveolar ridge augmentation, because it might be difficult to achieve a mechanical stability of implants at the implantation sites [585]. In addition, presence of calcium pyrophosphate impurity might be the reason for inflammation [588]. Additional details on inflammatory cell responses to CaPO$_4$ can be found in a special review on this topic [589].

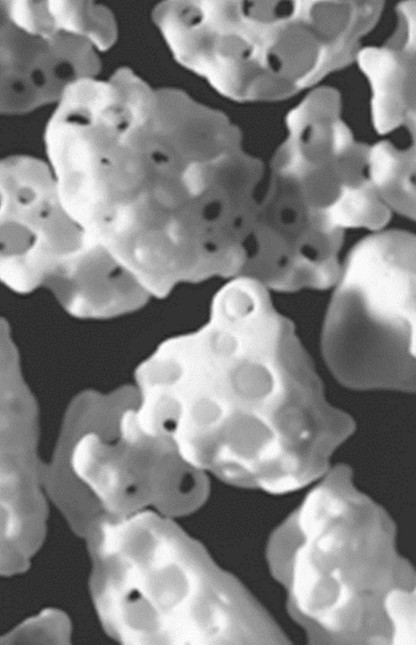

**Figure 11.** Rounded β-TCP granules 2.6–4.8 mm in size, providing no sharp edges for combination with bone cement. Reprinted from Ref. [595] with permission.

### 5.2. Osteoinduction

Until recently, it was generally considered that, alone, no type of synthetic bioceramic possessed either osteogenic (osteogenesis is the process of laying down new bone material by osteoblasts [596]) or osteoinductive (the property of the material to induce bone formation de novo or ectopically (i.e., in non-bone-forming sites) [596]) properties, and they demonstrated a minimal immediate structural support. However, a number of reports have already shown the osteoinductive properties of certain types of $CaPO_4$ bioceramics [544,566,597–605], and the amount of such publications is rapidly increasing. For example, bone formation was found to occur in dog muscle inside porous $CaPO_4$ with surface microporosity, while bone was not observed on the surface of dense bioceramics [601]. Furthermore, implantation of porous β-TCP bioceramics appeared to induce bone formation in soft tissues of dogs, while no bone formation was detected in any α-TCP implants [598]. More to the point, titanium implants coated with a microporous layer of OCP were found to induce ectopic bone formation in goat muscles, while a smooth layer of carbonated apatite on the same implants was not able to induce bone formation there [599,600]. In another study, β-TCP powder, biphasic (HA + β-TCP) powder, and intact biphasic (HA + β-TCP) rods were implanted into leg muscles of mice and dorsal muscles of rabbits [606]. One month and three months after implantation, samples were harvested for biological and histological analysis. New bone tissues were observed in 10 of 10 samples for β-TCP powder, 3 of 10 samples for biphasic powder, and 9 of 10 samples for intact biphasic rods at the third month in mice, but not in rabbits. The authors concluded that the chemical composition was the prerequisite in osteoinduction, while porosity contributed to more bone formation [606]. Therefore, researchers had already discovered the methods to prepare osteoinductive $CaPO_4$ bioceramics.

Unfortunately, the underlying mechanism(s) leading to bone induction by synthetic materials remains largely unknown. Nevertheless, besides the specific genetic factors [604] and chosen animals [606], the dissolution/precipitation behavior of $CaPO_4$ [607], their particle size [608,609], microporosity [572,603,610–614], physicochemical properties [601,603], composition [606], the specific surface area [614], and nanostructure [605], as well as the surface topography and geometry [602,615–621] have been pointed out as the relevant parameters [617]. A positive effect of increased microporosity on the ectopic bone formation could be both direct and indirect. Firstly, an increased microporosity is directly related to the changes in surface topography, i.e., it increases surface roughness, which affects the cellular differentiation [619]. Secondly, an increased microporosity indirectly means a larger surface that is exposed to the body fluids, leading to elevated dissolution/precipitation phenomena, as compared to non-microporous surfaces. In addition, other hypotheses are also available, namely, Reddi explained the apparent osteoinductive properties as an ability of particular bioceramics to concentrate bone growth factors, which are circulating in biological fluids, and those growth factors induce bone formation [615]. Other researchers proposed a similar hypothesis, that the intrinsic osteoinduction by $CaPO_4$ bioceramics is a result of adsorption of osteoinductive substances on their surface [602]. Moreover, Ripamonti [616] and Kuboki et al. [617] independently postulated that the geometry of $CaPO_4$ bioceramics is a critical parameter in bone induction. Specifically, bone induction by $CaPO_4$ was never observed on flat bioceramic surfaces. All osteoinductive cases were observed on either porous structures or structures that contained well-defined concavities. Furthermore, bone formation was never observed on the peripheries of porous implants and was always found inside the pores or concavities, aligning the surface [180]. Some researchers speculated that a low oxygen tension in the central region of implants might provoke a dedifferentiation of pericytes from blood microvessels into osteoblasts [622]. Finally, yet importantly, both nanostructured rough surfaces and a surface charge on implants were found to cause an asymmetrical division of the stem cells into osteoblasts, which is important for osteoinduction [613]. Additional details on this topic are available in the literature [623].

Nevertheless, to finalize this topic, it is worth citing a conclusion made by Boyan and Schwartz [624]: "Synthetic materials are presently used routinely as osteoconductive bone graft substitutes, but before purely synthetic materials can be used to treat bone defects in humans where an osteoinductive agent is required, a more complete appreciation of the biology of bone regeneration is needed. An understanding is needed of how synthetic materials modulate the migration, attachment, proliferation and differentiation of mesenchymal stem cells, how cells on the surface of a material affect other progenitor cells in the peri-implant tissue, how vascular progenitors can be recruited and a neovasculature maintained, and how remodeling of newly formed bone can be controlled." (p. 9).

*5.3. Biodegradation*

Shortly after implantation, a healing process is initiated by compositional changes of the surrounding bio-fluids and adsorption of biomolecules. Following this, various types of cells reach the $CaPO_4$ surface, and the adsorbed layer dictates the ways the cells respond. Further, a biodegradation (which can be envisioned as an in vivo process by which an implanted material breaks down into either simpler components or components of the smaller dimensions) of the implanted $CaPO_4$ bioceramics begins. This process can occur by three possible ways: (1) physical: due to abrasion, fracture and/or disintegration; (2) chemical: due to physicochemical dissolution of the implanted phases of $CaPO_4$ with a possibility of phase transformations into other phases of $CaPO_4$, as well as their precipitation; and (3) biological: due to cellular activity (so called, bioresorption). In biological systems, all these processes take place simultaneously and/or in competition with each other. For example, authors of interesting in vivo studies on a rat calvarial repair model showed that HA bioceramics degraded first, followed by diffusion of the degraded product, which was reconstructed to form new HA to repair the bone defect [625].

Since the existing $CaPO_4$ are differentiated by Ca/P ratio, basicity/acidity, and solubility (Table 1), in the first instance, their degradation kinetics and mechanisms depend on the chosen type of $CaPO_4$ [626,627]. Given the fact that dissolution is a physical chemistry process, it is controlled by some factors, such as $CaPO_4$ solubility, surface area to volume ratio, local acidity, fluid convection, and temperature. For HA and FA, the dissolution mechanism in acids has been described by a sequence of four successive chemical equations, in which several other $CaPO_4$, such as TCP, DCPD/DCPA and MCPM/MCPA, appear as virtual intermediate phases [628,629].

With a few exceptions, dissolution rates of $CaPO_4$ are inversely proportional to the Ca/P ratio (except for TTCP), phase purity, and crystalline size, and they are also directly related to both the porosity and the surface area. In addition, phase transformations might occur with DCPA, DCPD, OCP, α-TCP, β-TCP, and ACP because they are unstable in aqueous environments under the physiological conditions [630]. Bioresorption is a biological process mediated by cells (mainly osteoclasts and, to a lesser extent, macrophages) [631,632]. In vitro, this process may be followed up by various techniques, such as a spherical instrumented indentation [633]. Bioresorption depends on the response of cells to their environment. Osteoclasts attach firmly to the implant and dissolve $CaPO_4$ by secreting an enzyme carbonic anhydrase or any other acid, leading to a local pH drop to ~4–5 [634]. Formation of multiple spine-like crystals at the exposed areas of β-TCP was discovered [635]. Furthermore, nanodimensional particles of $CaPO_4$ can also be phagocytosed by cells, i.e., they are incorporated into cytoplasm and thereafter dissolved by acid attack and/or enzymatic processes [636]. A study is available [637] in which a comparison was made between the solubility and osteoclastic resorbability of three types of $CaPO_4$ (DCPA, ACP, and HA) + β-calcium pyrophosphate (β-CPP) powders having the monodisperse particle size distributions. The authors discovered that with the exception of β-CPP, the difference in solubility among different calcium phosphates became neither mitigated nor reversed but augmented in the resorptive osteoclastic milieu. Namely, DCPA (the phase with the highest solubility) was resorbed more intensely than any other calcium phosphate, whereas HA (the phase with the lowest solubility) was resorbed the least. B-CPP became retained inside

the cells for the longest period of time, indicating hindered digestion of only this particular type of calcium phosphate. Genesis of osteoclasts was found to be mildly hindered in the presence of HA, ACP, and DCPA, but not in the presence of β-CPP. HA appeared to be the most viable compound with respect to the mitochondrial succinic dehydrogenase activity. The authors concluded that chemistry did have a direct effect on biology, while biology neither overrode nor reversed the chemical propensities of calcium phosphates with which it interacted, but rather augmented and took a direct advantage of them [637]. Similar conclusions on both the resorbability and dissolution behavior of OCP, β-TCP, and HA [630], as well as β-TCP, BCP (HA + β-TCP), and HA [638], were made by other researchers. In addition, in vivo biodegradation of MCPA was found to be faster than that of bovine HA [639]. Thus, one can conclude that in vivo biodegradation kinetics of $CaPO_4$ seem to correlate well with their solubility. Nevertheless, one must keep in mind that this is a very complicated combination of various nonequilibrium processes, occurring simultaneously and/or in competition with each other [640].

Strictly speaking, the processes that happen in vitro do not necessarily represent the ones occurring in vivo and vice versa; nevertheless, in vitro experiments are widely performed. Usually, an in vitro biodegradation of $CaPO_4$ bioceramics is simulated by suspending the material in a slightly acidic (pH~4) buffer and monitoring the release of major ions with time [627,641–644]. An acidic buffer, to some extent, mimics the acidic environment during osteoclastic activity. The authors of one study reviewed the available literature on acellular in vitro resorption of $CaPO_4$ bioceramics and found the following [645]: "The materials were certainly processed under different conditions, but this dispersion of data is also due to the large variety of tests performed. In fact, each work differs from the others in the type of sample (i.e., composition, shape, porosity, dimension.), of immersion condition (i.e., kind of solution, quantity, stirring, refresh.) and of performed analysis (i.e., microstructural, physicochemical, mechanical) and testing conditions. However, all these aspects can affect the final results." (p. 912). Further, the authors of that paper performed in vitro resorption of DCPD and β-TCP samples in TRIS and PBS solutions for different times with or without refresh of the medium and demonstrated the importance of choosing the appropriate immersion conditions according to the phenomenon being investigated (i.e., $CaPO_4$ dissolution, precipitation of new phases, etc.) [645].

For example, in vivo behavior of porous β-TCP bioceramics prepared from rod-shaped particles and those prepared from non-rod-shaped particles in the rabbit femur was compared. Although the porosities of both types of β-TCP bioceramics were almost the same, a more active osteogenesis was preserved in the region where rod-shaped bioceramics were implanted [646]. Furthermore, the dimensions of both the particles [608] and the surface microstructure [607] were found to influence the osteoinductive potential of $CaPO_4$ bioceramics. These results implied that the microstructure affected the activity of bone cells and subsequent bone replacement.

In addition, a quantitative and fast method was developed to measure the chemical changes occurring within the pores of β-TCP granules incubated in a simulated body fluid [647]. A factorial design of experiments revealed that the particle size, specific surface area, microporosity, and purity of the β-TCP granules influenced the chemical composition of the solution. Large pH, calcium, and phosphate concentration changes were observed inside the granules and lasted for several days. The kinetics and magnitude of these changes (up to 2 pH units) largely depended on the processing and properties of the granules. Small particles, low sintering temperature, high microporosity, and the presence of HA impurity magnified the intensity and duration of pH, calcium, and phosphate variations [647].

Regarding in vivo studies, terbium (Tb)-doped uniform nanodimensional CDHA crystals were implanted into bone tissue and compared with those of native bone apatite. The comparisons demonstrated an occurrence of compositional and structural alterations of the implanted CDHA crystals and their gradual degradation during bone reconstruction. They also revealed notable differences between implanted Tb-doped CDHA and bone apatite crystals in dimensions, distribution pattern, and state of existence in bone tissue.

The authors concluded that although synthetic nanodimensional CDHA crystals could osteointegrate with bone tissue, they still seemed to be treated as foreign material and thus were gradually degraded [648].

The experimental results demonstrated that both the dissolution kinetics and in vivo biodegradation of biologically relevant $CaPO_4$ proceed in the following decreasing order: β-TCP > bovine bone apatite (unsintered) > bovine bone apatite (sintered) > coralline HA > HA. In the case of biphasic (HA + TCP), triphasic, and multiphasic $CaPO_4$ formulations, the biodegradation kinetics depend on the HA/TCP ratio: the higher the ratio, the lower the degradation rate. Similarly, the in vivo degradation rate of biphasic TCP (α-TCP + β-TCP) bioceramics appeared to be lower than that of α-TCP and higher than that of β-TCP bioceramics, respectively [93]. Furthermore, incorporation of doping ions can either increase (e.g., $CO_3^{2-}$, $Mg^{2+}$, or $Sr^{2+}$) or decrease (e.g., $F^-$) the solubility (therefore, biodegradability) of CDHA and HA. Contrarily to apatites, solubility of β-TCP is decreased by incorporation of either $Mg^{2+}$ or $Zn^{2+}$ ions [544]. Here, one should remember that ion-substituted $CaPO_4$ are not considered in this review; the interested readers are advised to read the original publications [17–41].

*5.4. Bioactivity*

Generally, bioactive materials interact with surrounding bone, resulting in formation of a chemical bond to this tissue (bone bonding). The bioactivity phenomenon is determined by both chemical factors, such as crystal phases and molecular structures of a biomaterial, and physical factors, such as surface roughness and porosity. Currently, it is agreed that the newly formed bone bonds directly to biomaterials through a carbonated CDHA layer precipitating at the bone/biomaterial interface. Strangely enough, a careful search of the literature resulted in just a few publications [544,623,649–651] where the bioactivity mechanism of $CaPO_4$ was briefly described. For example, the chemical changes occurring after exposure of a synthetic HA bioceramic to both in vivo (implantation in human) and in vitro (cell culture) conditions were studied. A small amount of HA was phagocytozed but the major remaining part behaved as a secondary nucleator, as evidenced by the appearance of a newly formed mineral [649]. In vivo, cellular activity (e.g., of macrophages or osteoclasts; however, this may depend on the cellular origin [652]) associated with an acidic environment was found to result in partial dissolution of $CaPO_4$, causing liberation of calcium and orthophosphate ions to the microenvironment. The liberated ions increased the local supersaturation degree of the surrounding biologic fluids, causing precipitation of nanosized crystals of biological apatite with simultaneous incorporating of various ions present in the fluids. Infrared spectroscopic analyses demonstrated that these nanodimensional crystals were intimately associated with bioorganic components (probably proteins), which might also have originated from the biologic fluids, such as serum [544]. However, in 2019, the concept of a local supersaturation degree that caused $CaPO_4$ precipitation was criticized: on the contrary, intrinsic osteoinduction was proposed to be the result of calcium and/or phosphate depletion (blood supply must be insufficient to maintain the physiological calcium and/or phosphate ion concentrations) [623].

Therefore, one should consider the bioactivity mechanism of other biomaterials, particularly of bioactive glasses—the concept introduced by Prof. Larry L. Hench [50,51]. The bonding mechanism of bioactive glasses to living tissues involves a sequence of 11 successive reaction steps (Figure 12), some of which comprise $CaPO_4$. The initial five steps occurring on the surface of bioactive glasses are "chemistry" only, while the remaining six steps belong to "biology" because the latter include colonization by osteoblasts, followed by proliferation and differentiation of the cells to form a new bone that has a mechanically strong bond to the implant surface. Therefore, in the case of bioactive glasses, the border between "dead" and "alive" is postulated between stages 5 and 6. According to Hench, all bioactive materials "form a bone-like apatite layer on their surfaces in the living body and bond to bone through this apatite layer. The formation of bone-like apatite on artificial material is induced by functional groups, such as Si–OH (in the case of biological glasses),

Ti–OH, Zr–OH, Nb–OH, Ta–OH, –COOH and –H$_2$PO$_4$ (in the case of other materials). These groups have specific structures revealing negatively charge and induce apatite formation via formations of an amorphous calcium compound, e.g., calcium silicate, calcium titanate and ACP" [50,51].

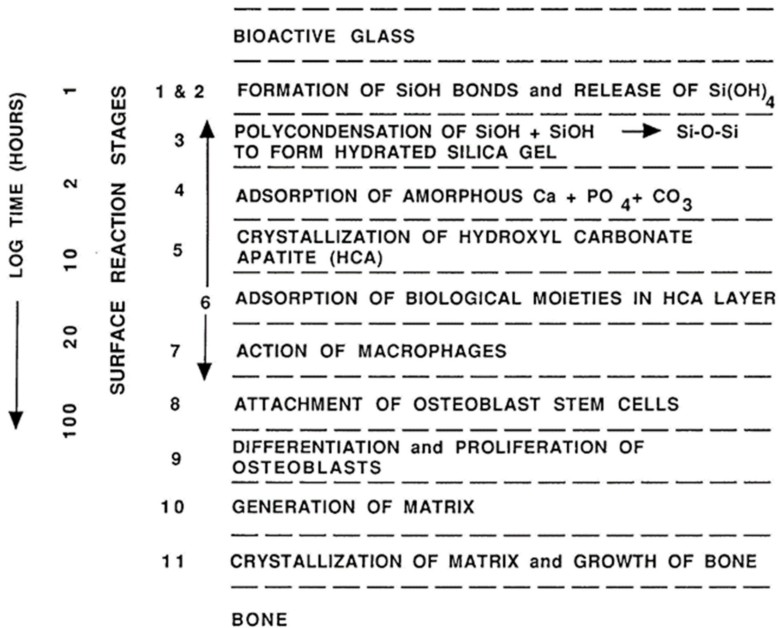

**Figure 12.** A sequence of interfacial reactions involved in forming a bond between tissue and bioactive ceramics. Reprinted from Refs. [50,51] with permission.

In addition, one should mention another set of 11 successive reaction steps for bonding mechanism of unspecified bioceramics, developed by Prof. Paul Ducheyne (Figure 13) [58]. One can see that the Ducheyne's model is rather similar to that proposed by Hench; however, there are noticeable differences between them. For example, Ducheyne mentions ion exchange and structural rearrangement at the bioceramic/tissue interface (stage 3), as well as on interdiffusion from the surface boundary layer into bioceramics (stage 4) and deposition with integration into the bioceramics (stage 7), which are absent in the Hench's model. On the other hand, Hench describes six biological stages (stages 6–11), while Ducheyne describes only four (stages 8–11). Both models were developed more than two decades ago and, to the best of my knowledge, remain unchanged since then. Presumably, both approaches have pro et contra of their own and, obviously, should be updated and/or revised. Furthermore, in the literature there are at least two other descriptions of the biological and cellular events occurring at the bone/implant interface [653,654]. Unfortunately, both of them comprise fewer stages. In 2010, one more hypothesis was proposed (Figure 14). For the first time, it describes reasonable surface transformations, happening with CaPO$_4$ bioceramics (in that case, HA) shortly after the implantation [651]. However, one must note that the schemes displayed in Figures 12–14 do not represent the real mechanisms, but are only descriptions of the observable events occurring at the CaPO$_4$ interface after implantation. Furthermore, many events occur simultaneously; therefore, none of the schemes should be considered in terms of the strict time sequences.

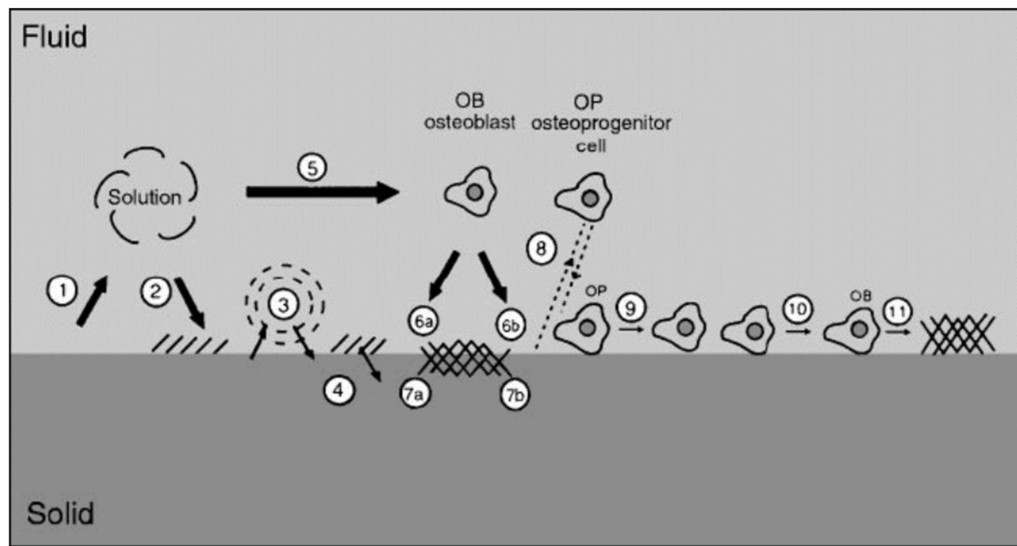

**Figure 13.** A schematic diagram representing the events which take place at the interface between bioceramics and the surrounding biological environment: (1) dissolution of bioceramics; (2) precipitation from solution onto bioceramics; (3) ion exchange and structural rearrangement at the bioceramic/tissue interface; (4) interdiffusion from the surface boundary layer into the bioceramics; (5) solution-mediated effects on cellular activity; (6) deposition of either the mineral phase (a) or the organic phase (b) without integration into the bioceramic surface; (7) deposition with integration into the bioceramics; (8) chemotaxis to the bioceramic surface; (9) cell attachment and proliferation; (10) cell differentiation; (11) extracellular matrix formation. All phenomena, collectively, lead to the gradual incorporation of a bioceramic implant into developing bone tissue. Reprinted from Ref. [58] with permission.

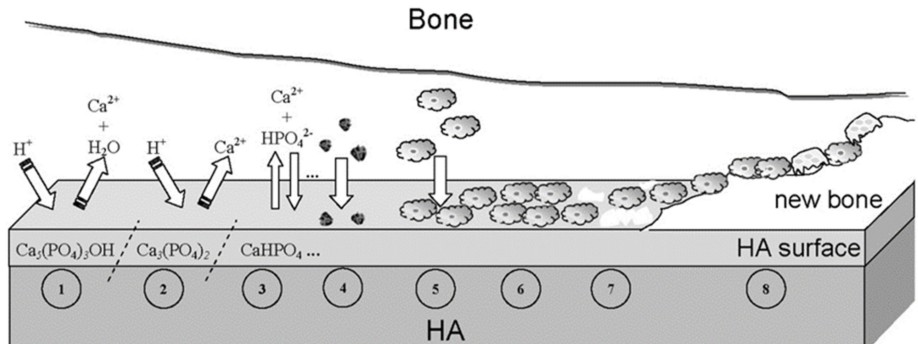

**Figure 14.** A schematic diagram representing the phenomena that occur on the HA surface after implantation: (1) beginning of the implant procedure, where a solubilization of the HA surface starts; (2) continuation of the solubilization of the HA surface; (3) the equilibrium between the physiological solutions and the modified surface of HA has been achieved (changes in the surface composition of HA do not mean that a new phase of DCPA or DCPD forms on the surface); (4) adsorption of proteins and/or other bioorganic compounds; (5) cell adhesion; (6) cell proliferation; (7) beginning of a new bone formation; (8) new bone has been formed. Reprinted from Ref. [651] with permission.

An important study on formation of $CaPO_4$ precipitates on various types of bioceramic surfaces in both simulated body fluid and rabbit muscle sites was performed [655]. The bioceramics were sintered porous solids, including bioglass, glass-ceramics, $\alpha$-TCP, $\beta$-TCP, and HA. The ability to induce $CaPO_4$ precipitation was compared among these types of bioceramics. The following conclusions were made: (1) OCP formation ubiquitously occurred on all types of bioceramic surfaces both in vitro and in vivo, except on $\beta$-TCP. (2) Apatite formation did not occur on every type of bioceramic surface; it was less likely to occur on the surfaces of HA and $\alpha$-TCP. (3) Precipitation of $CaPO_4$ on the bioceramic

surfaces was more difficult in vivo than in vitro. (4) Differences in $CaPO_4$ precipitation among the bioceramic surfaces were less noticeable in vitro than that in vivo. (5) $\beta$-TCP bioceramics showed poor ability of $CaPO_4$ precipitation both in vitro and in vivo [655]. These findings clearly revealed that apatite formation in the physiological environments could not be confirmed as the common feature of bioceramics. Nevertheless, for want of anything better, currently, the bioactivity mechanism of $CaPO_4$ bioceramics should be described by a reasonable combination of Figures 12–14, e.g., by updating the Ducheyne's and Hench's models with the three initial stages taken from Figure 14. Additional details on this topic are available in the literature [656].

Interestingly, bioactivity of HA bioceramics might be enhanced by a high-energy ion irradiation [657]. The effect was attributed to formation of a unique 3D macroporous apatite layer of decreased crystallinity and crystal size on the irradiated surfaces. Obviously, to obtain further insights into the bioactivity phenomenon, the atomic and molecular processes occurring at the bioceramic surface in aqueous solutions and their effects on the relevant reaction pathways of cells and tissues must be elucidated in more details.

### 5.5. Cellular Response

Fixation of any implants in the body is a complex dynamic process that remodels the interface between the implants and living tissues at all dimensional levels, from the molecular up to the cell and tissue morphology level, and at all time scales, from the first second up to several years after implantation. Immediately following the implantation, a space filled with biological fluids appears next to the implant surface. With time, cells are adsorbed at the implant surface that will give rise to their proliferation and differentiation towards bone cells, followed by revascularization and eventual gap closing. Ideally, a strong bond is formed between the implants and surrounding tissues [53]. An interesting study on the interfacial interactions between calcined HA and substrates was performed [658], where the interested readers are referred for further details.

The aforementioned paragraph clearly demonstrates the importance of studies on cellular responses to $CaPO_4$ bioceramics. Such investigations have been performed extensively for several decades [589,659–671]. For example, bioceramic discs made of seven different types of $CaPO_4$ (TTCP, HA, carbonate apatite, $\beta$-TCP, $\alpha$-TCP, OCP, and DCPD) were incubated in osteoclastic cell cultures for 2 days. In all cases, similar cell morphologies and good cell viability were observed; however, different levels of resorbability of various types of $CaPO_4$ were detected [661]. Similar results were found for fluoridated HA coatings [663]. Chemical composition of $CaPO_4$, which contributed to pH changes, and concentration of calcium ions in the medium were found to make up particularly significant factors for cellular responses; moreover, it was proved that the number of material types represented a further important aspect [670]. Experiments performed with human osteoblasts revealed that nanostructured bioceramics prepared from nanosized HA showed significant enhancement in mineralization compared to microstructured HA bioceramics [662]. In addition, the influence of lengths and surface areas of rod-shaped HA on cellular response were studied. Again, similar cell morphologies and good cell viability were observed; however, it was concluded that high surface area could increase cell–particle interaction [665]. Nevertheless, another study with cellular response to rod-shaped HA bioceramics revealed that some types of crystals might trigger a severe inflammatory response [666]. In addition, $CaPO_4$-based sealers appeared to show fewer cytotoxicity and inflammatory mediators compared with other sealers [664]. More examples are available in the literature [589,659–671].

Cellular biodegradation of $CaPO_4$ bioceramics is known to depend on its phases. For example, a higher solubility of $\beta$-TCP was shown to prevent L-929 fibroblast cell adhesion, thereby leading to damage and rupture of the cells [672]. A mouse ectopic model study indicated the maximal bone growth for the 80:20 $\beta$-TCP:HA biphasic formulations preloaded with human mesenchymal stem cells when compared to other $CaPO_4$ [673]. The effects of substrate microstructure and crystallinity have been corroborated with an

in vivo rabbit femur model, where rod-like crystalline β-TCP was reported to enhance osteogenesis when compared to non-rod-like crystalline β-TCP [649]. Additionally, using a dog mandibular defect model, a higher bone formation on a scaffold surface coated by nanodimensional HA was observed when compared to that coated by a microdimensional HA [674]. Furthermore, studies revealed a stronger stress signaling response by osteoblast precursor cells in 3D scaffolds when compared to 2D surfaces [675].

Mesenchymal stem cells are one of the most attractive cellular lines for application as bone grafts [676,677]. Early investigations by Okumura et al. indicated an adhesion, proliferation, and differentiation, which ultimately became new bone and integrated with porous HA bioceramics [660]. Later, a sustained coculture of endothelial cells and osteoblasts on HA scaffolds for up to 6 weeks was demonstrated [678]. Furthermore, a release of factors by endothelial and osteoblast cells in coculture-supported proliferation and differentiation was suggested to ultimately result in microcapillary-like vessel formation and supported a neo-tissue growth within the scaffold [479]. More to the point, investigation of rat calvaria osteoblasts cultured on transparent HA bioceramics, as well as the analysis of osteogenic-induced human bone marrow stromal cells at different time points of culturing, indicated a good cytocompatibility of HA bioceramics and revealed favorable cell proliferation [424]. Positive results for other types of cells were obtained in other studies [187,423,443,444,679–681]. In addition, $CaPO_4$ are used for cell transfections [682].

Interestingly, HA scaffolds with marrow stromal cells in a perfused environment were reported to result in ~85% increase in mean core strength, a ~130% increase in failure energy, and a ~355% increase in post-failure strength. The increase in mineral quantity and promotion of the uniform mineral distribution in that study was suggested to be attributed to the perfusion effect [560]. Additionally, other investigators indicated mechanical properties increasing for other $CaPO_4$ scaffolds after induced osteogenesis [559,562].

To finalize this subsection, one should note the recent developments to influence the cellular response. First, to facilitate interactions with cells, the $CaPO_4$ surfaces could be functionalized [683–686]. Second, it appears that crystals of biological apatite of calcified tissues exhibit different orientations depending on the tissue; namely, in vertebrate bones and tooth enamel surfaces, the respective *a, b*-planes and *c*-planes of the apatite crystals are preferentially exposed. Therefore, ideally, this should be taken into account in artificial bone grafts. Recently, a novel process to fabricate dense HA bioceramics with highly preferred orientation to the *a,b*-plane was developed. The results revealed that increasing the *a,b*-plane orientation degree shifted the surface charge from negative to positive and decreased the surface wettability with simultaneous decreasing of cell attachment efficiency [687–689]. The latter finding resulted in further developments on preparation of oriented $CaPO_4$ compounds [690–692].

Finally, to conclude the entire *Biological Properties and in vivo Behavior* section, let me quote several sentences from Ref. [265]: "Variations in surface chemistry resulting from variable thermal processing conditions of otherwise identical samples might thus explain inconsistencies in biological behavior reported in the literature. For instance, β-TCP has been reported to be both bioactive [693] and non-bioactive [655], non-osteoinductive [694] and highly osteoinductive [695,696], highly resorbable [694,697] and poorly resorbable [697]. Authors have related this dichotomous behavior with the effect of sintering temperature on specific surface area, bulk composition, and scaffold or pore topography." (p. 6096). In addition, simple thermal treatment at 500 °C was found to reduce body reactions to irregular α- and β-TCP granules as foreign bodies, due to a partial evaporation of phosphate species during thermal treatment [698]. Thus, there are still many uncertainties in our understanding of the biological properties of $CaPO_4$ bioceramics.

## 6. Biomedical Applications

Since Levitt et al. described a method of preparing FA bioceramics and suggested their possible use in medical applications in 1969 [699], $CaPO_4$ bioceramics have been widely tested for clinical applications. Namely, over 400 forms, compositions, and trademarks

(Table 3) are currently either in use or under consideration in many areas of orthopedics and dentistry [700], with even more in development. In addition, various formulations containing demineralized bone matrix (commonly abbreviated as DBM) are produced for bone grafting. For example, bulk materials, available in dense and porous forms, are used for alveolar ridge augmentation, immediate tooth replacement, and maxillofacial reconstruction [4,701]. Other examples comprise burr-hole buttons [702,703], cosmetic (nonfunctional) eye replacements such as Bio-Eye® [704–709], increment of the hearing ossicles [710–712], and spine fusion [713–716], as well as repair of bone [118,717,718], craniofacial [719], and dental [720] defects. In order to permit growth of new bone into defects, a suitable bioresorbable material should fill these defects. Otherwise, ingrowth of fibrous tissue might prevent bone formation within the defects.

**Table 3.** Registered commercial trademarks (current and past) of CaPO₄-based bioceramics and biomaterials.

| Calcium Orthophosphate | Trade Name and Producer (When Available) |
| --- | --- |
| CDHA | Calcibon (Zimmer Biomet, IN, USA) |
| | Cementek (Teknimed, France) |
| | CHT Ceramic Hydroxyapatite (Bio-Rad, CA, USA) |
| | nanoXIM (Fluidinova, Portugal) |
| | OsteoGen (Impladent, NY, USA) |
| | without trade name (Himed, NY, USA) |
| HA | Actifuse (ApaTech, UK) |
| | Alveograf (Cooke-Waite Laboratories, USA) |
| | Apaceram (HOYA Technosurgical, Japan) |
| | Apafill-G (Habana, Cuba) |
| | ApaPore (ApaTech, UK) |
| | BABI-HAP (Berkeley Advanced Biomaterials, CA, USA) |
| | Bio-Eye (Integrated Orbital Implants, CA, USA) |
| | BIOGAP (Connectbiopharm, Russia) |
| | BioGraft (IFGL BIO CERAMICS, India) |
| | Bioroc (Depuy Bioland, France) |
| | Blue Bone (Regener Biomateriais, Brazil) |
| | Boneceram (Sumitomo Osaka Cement, Japan) |
| | Bonefil (Pentax, Japan) |
| | BoneSource (Stryker Orthopaedics, NJ, USA) |
| | Bonetite (Pentax, Japan) |
| | Bonfil (Mitsubishi Materials, Japan) |
| | Bongros-HA (Daewoong Pharmaceutical, Korea) |
| | CAFOS DT (Chemische Fabrik Budenheim, Germany) |
| | Calcitite (Sulzer Calcitek, CA, USA) |
| | CAMCERAM HA (CAM Implants, Netherlands) |
| | CAPTAL (Plasma Biotal, UK) |
| | CELLYARD (HOYA Technosurgical, Japan) |
| | Cerapatite (Ceraver, France) |
| | Ceros HA (Mathys, Switzerland) |

**Table 3.** *Cont.*

| Calcium Orthophosphate | Trade Name and Producer (When Available) |
| --- | --- |
| | CHT Ceramic Hydroxyapatite (Bio-Rad, CA, USA) |
| | Durapatite (unknown producer) |
| | ENGIpore (JRI Orthopaedics, UK) |
| | G-Bone (Surgiwear, India) |
| | GranuMas (GranuLab, Malaysia) |
| | HA BIOCER (CHEMA – ELEKTROMET, Poland) |
| | HA$^{nano}$ Surface (Promimic, Sweden) |
| | HAP-91 (JHS Biomateriais, Brazil) |
| | HAP-99 (Polystom, Russia) |
| | HAP–Bionnovation (Bionnovation, Brazil) |
| | IngeniOs HA (Zimmer Dental, CA, USA) |
| | Micro Crystalline Hydroxyapatite Complex (MCHC) (Clarion Pharmaceutical, India) |
| | nanoXIM (Fluidinova, Portugal) |
| | Neobone (Covalent Materials, Japan) |
| | Osbone (Curasan, Germany) |
| | OsproLife HA (Lincotek Medical, Italy) |
| | Ossein Hydroxyapatite (Clarion Pharmaceutical, India) |
| | OssaBase-HA (Lasak, Czech Republic) |
| | Ostegraf (Ceramed, CO, USA) |
| | Ostim (Heraeus Kulzer, Germany) |
| | Ovis Bone HA (DENTIS, Korea) |
| | Periograf (Cooke-Waite Laboratories, USA) |
| | PermaOS (Mathys, Switzerland) |
| | PRINT3D Hydroxyapatite (Prodways, France) |
| | Pro Osteon (Zimmer Biomet, IN, USA) |
| | PurAtite (PremierBiomaterials, Ireland) |
| | REGENOS (Kuraray, Japan) |
| | SHAp (SofSera, Japan) |
| | Synatite (SBM, France) |
| | Synthacer (KARL STORZ Recon, Germany) |
| | Theriridge (Therics, OH, USA) |
| | without trade name (Cam Bioceramics, Netherlands) |
| | without trade name (CaP Biomaterials, WI, USA) |
| | without trade name (DinganTec, China) |
| | without trade name (Ensail Beijing, China) |
| | without trade name (Himed, NY, USA) |
| | without trade name (MedicalGroup, France) |
| | without trade name (SANGI, Japan) |
| | without trade name (Shanghai Rebone Biomaterials, China) |
| | without trade name (SigmaGraft, CA, USA) |

**Table 3.** *Cont.*

| Calcium Orthophosphate | Trade Name and Producer (When Available) |
| --- | --- |
| | without trade name (SkySpring Nanomaterials, TX, USA) |
| | without trade name (SofSera, Japan) |
| | without trade name (Taihei Chemical Industrial, Japan) |
| | without trade name (Xpand Biotechnology, Netherlands) |
| Mg-HA | SINTlife (JRI Orthopaedics, UK) |
| HA powder suspended in water | Ostibone (FH Orthopedics, France) |
| | NANOSTIM (Medtronic Sofamor Danek, TN, USA) |
| | n-IBS (Bioceramed, Portugal) |
| | Skelifil (Osteotec, UK) |
| HA embedded or suspended in a gel | Bio-Gel HT hydroxyapatite (Bio-Rad, CA, USA) |
| | Coaptite (Boston Scientific, MA, USA) |
| | Facetem (Daewoong, Korea) |
| | NanoBone (Artoss, Germany) |
| | Nanogel (Teknimed, France) |
| | Radiesse (Merz Aesthetics, Germany) |
| | Renú Calcium Hydroxylapatite Implant (Cytophil, WI, USA) |
| HA/collagen, CDHA/collagen and/or carbonate apatite/collagen | AUGMATRIX (Wright Medical Technology, TN, USA) |
| | Bioimplant (Connectbiopharm, Russia) |
| | Bio-Oss Collagen (Geitslich, Switzerland) |
| | Boneject (Koken, Japan) |
| | COL.HAP-91 (JHS Biomateriais, Brazil) |
| | Collagraft (Zimmer and Collagen Corporation, USA) |
| | CollaOss (SK Bioland, Korea) |
| | CollapAn (Intermedapatite, Russia) |
| | COLLAPAT (Symatese, France) |
| | DualPor collagen (OssGen, Korea) |
| | G-Graft (Surgiwear, India) |
| | HAPCOL (Polystom, Russia) |
| | Healos (DePuy Spine, USA) |
| | LitAr (LitAr, Russia) |
| | Ossbone Collagen (SK Bioland, Korea) |
| | OssFill (Sewon Cellontech, Korea) |
| | OssiMend (Collagen Matrix, NJ, USA) |
| | Osteomatrix (Connectbiopharm, Russia) |
| | OsteoTape (Impladent, NY, USA) |
| | ReFit (HOYA Technosurgical, Japan |
| | RegenOss (JRI Orthopaedics, UK) |
| | RegenerOss Synthetic (Zimmer Dental, CA, USA) |
| | Straumann XenoFlex (Straumann, Switzerland) |

**Table 3.** *Cont.*

| Calcium Orthophosphate | Trade Name and Producer (When Available) |
|---|---|
| HA/sodium alginate | Bialgin (Biomed, Russia) |
| HA/poly-L-lactic acid | Biosteon (Biocomposites, UK) |
| | ReOss (ReOss, Germany) |
| | OSTEOTRANS MX (Teijin Medical Technologies, Japan) |
| | SuperFIXSORB30 (Takiron, Japan) |
| HA/polyethylene | HAPEX (Gyrus, TN, USA) |
| HA/CaSO$_4$ | BioWrist Bone Void Filler (Skeletal Kinetics, CA, USA) |
| | Bond Apatite (Augma Biomaterials, NJ, USA) |
| | Hapset (LifeCore, MN, USA) |
| | PerOssal (aap Implantate, Germany) |
| HA/CaSO$_4$ powders suspended in a liquid | CERAMENT (BONESUPPORT, Sweden) |
| Coralline HA | Biocoral (Bio Coral Calcium Bone, France) |
| | BoneMedik-S (Meta Biomed, Korea) |
| | Interpore (Interpore, CA, USA) |
| | ProOsteon (Interpore, CA, USA) |
| Carbonate apatite | Cytrans (GC, Japan) |
| | Norian SRS (Norian, CA, USA) |
| Algae-derived HA | Algipore (AlgOss Biotechnologies, Austria) |
| | Algisorb (AlgOss Biotechnologies, Austria) |
| | FRIOS Algipore (DENTSPLY Implants, Sweden) |
| | SIC nature graft (AlgOss Biotechnologies, Austria) |
| HA/glass | Bonelike (unknwn producer) |
| Bovine bone (unsintered) | Unilab Surgibone (Unilab, NJ, USA) |
| Bovine bone (unsintered) + polymer | Alpha-Bio's Graft (Alpha-Bio Tec, Israel) |
| | C-Graft Putty (unknwn producer) |
| Bovine bone apatite (unsintered) | Apatos (OsteoBiol, Italy) |
| | Bio-Oss (Geistlich Biomaterials, Switzerland) |
| | Bonefill (Bionnovation, Brazil). |
| | CANCELLO-PURE (Wright Medical Technology, TN, USA) |
| | CenoBone (Tissue Regeneration Corporation, Iran) |
| | CopiOs Cancellous Particulate Xenograft (Zimmer, IN, USA) |
| | GenOs (OsteoBiol, Italy) |
| | InterOss (SigmaGraft, CA, USA) |
| | Laddec (Ost-Developpement, France) |
| | Lubboc (Ost-Developpement, France) |
| | MatrixCellect (Curasan, Germany) |
| | Mega-Oss Bovine (Megagen Implant, Korea) |
| | Orthoss (Geitslich, Switzerland) |
| | OssiGuide (Collagen Matrix, NJ, USA) |
| | Oxbone (Bioland biomateriaux, France) |

**Table 3.** *Cont.*

| Calcium Orthophosphate | Trade Name and Producer (When Available) |
| --- | --- |
| | Straumann XenoGraft (Straumann, Switzerland) |
| | Surgibone (Surgibon, Ecuador) |
| | Tutobone (Tutogen Medical, Germany) |
| | Tutofix (Tutogen Medical, Germany) |
| | Tutoplast (Tutogen Medical, Germany) |
| | without trade name (MedicalGroup, France) |
| Porcine bone apatite (unsintered) | A-OSS (Osstem Implant, Korea) |
| | GEM Bone Graft (Lynch Biologics, USA) |
| | Gen-Os (OsteoBiol, Italy) |
| | MatrixOss (Collagen Matrix, NJ, USA) |
| | OsteoBiol (OsteoBiol, Italy) |
| | Symbios Xenograft (DENTSPLY Implants, Sweden) |
| | THE Graft (Purgo Biologics, Korea) |
| Equine bone apatite (unsintered) | BIO-GEN (BioTECK, Italy) |
| | Sp-Block (OsteoBiol, Italy) |
| Bovine bone apatite (sintered) | 4Bone XBM (MIS Implants, Israel) |
| | BonAP (unknown producer) |
| | Cerabone (aap Implantate, Germany and botiss, Germany) |
| | Endobon (Merck, Germany) |
| | GenoxInorgânico (Baumer, SP, Brazil) |
| | Iceberg oss (Global Medical Implants, Spain) |
| | Navigraft (Zimmer Dental, USA) |
| | Osteograf (Ceramed, CO, USA) |
| | OVIS XENO (DENTIS, Korea) |
| | PepGen P-15 (DENTSPLY Implants, Sweden) |
| | Pyrost (Osteo AG, Germany) |
| | Sinbone (Purzer Pharmaceutical, Taiwan) |
| | SynOss (Collagen Matrix, NJ, USA) |
| | Straumann cerabone (Straumann, Switzerland) |
| Human bone allograft | ALLOPURE (Wright Medical Technology, TN, USA) |
| | Allosorb (Curasan, Germany) |
| | CancellOss (Impladent, NY, USA) |
| | CurOss (Impladent, NY, USA) |
| | J Bone Block (Impladent, NY, USA) |
| | maxgraft (botiss, Germany) |
| | Mega-Oss (Megagen Implant, Korea) |
| | NonDemin (Impladent, NY, USA) |
| | Osnatal (aap Implantate, Germany) |
| | OsteoDemin (Impladent, NY, USA) |
| | OsteoWrap (Curasan, Germany) |
| | OVIS ALLO (DENTIS, Korea) |

**Table 3.** *Cont.*

| Calcium Orthophosphate | Trade Name and Producer (When Available) |
|---|---|
| | PentOS OI (Citagenix, QC, Canada) |
| | RAPTOS (Citagenix, QC, Canada) |
| | Straumann AlloGraft (Straumann, Switzerland) |
| | TenFUSE (Wright Medical Technology, TN, USA) |
| α-TCP | BioBase (Biovision, Germany) |
| | Tetrabone (unknown producer) |
| | without trade name (Cam Bioceramics, Netherlands) |
| | without trade name (DinganTec, China) |
| | without trade name (Ensail Beijing, China) |
| | without trade name (Himed, NY, USA) |
| | without trade name (InnoTERE, Germany) |
| | without trade name (PremierBiomaterials, Ireland) |
| | without trade name (Taihei Chemical Industrial, Japan) |
| β-TCP | AdboneTCP (Medbone Medical Devices, Portugal) |
| | AFFINOS (Kuraray, Japan) |
| | Allogran-R (Biocomposites, UK) |
| | Antartik TCP (MedicalBiomat, France) |
| | ArrowBone (Brain Base Corporation, Japan) |
| | AttraX scaffold (NuVasive, CA, USA) |
| | BABI-TCP (Berkeley Advanced Biomaterials, CA, USA) |
| | Betabase (Biovision, Germany) |
| | BioGraft (IFGL BIO CERAMICS, India) |
| | Bioresorb (Sybron Implant Solutions, Germany) |
| | Biosorb (SBM, France) |
| | Bi-Ostetic (Berkeley Advanced Biomaterials, CA, USA) |
| | Bonegraft (Bonegraft biomaterials, Turkey) |
| | BoneSigma TCP (SigmaGraft, CA, USA) |
| | C 13-09 (Chemische Fabrik Budenheim, Germany) |
| | Calc-i-oss classic (Degradable Solutions, Switzerland) |
| | Calciresorb (Ceraver, France) |
| | CAMCERAM TCP (CAM Implants, Netherlands) |
| | CAPTAL β-TCP (Plasma Biotal, UK) |
| | CELLPLEX (Wright Medical Technology, TN, USA) |
| | Cerasorb (Curasan, Germany) |
| | Ceros TCP (Mathys, Switzerland) |
| | ChronOS (Synthes, PA, USA) |
| | Cidemarec (KERAMAT, Spain) |
| | Conduit (DePuy Spine, USA) |
| | cyclOS (Mathys, Switzerland) |
| | ExcelOs (BioAlpha, Korea) |
| | GenerOs (Berkeley Advanced Biomaterials, CA, USA) |

**Table 3.** *Cont.*

| Calcium Orthophosphate | Trade Name and Producer (When Available) |
| --- | --- |
| | HT BIOCER (CHEMA – ELEKTROMET, Poland) |
| | Iceberg TCP (Global Medical Implants, Spain) |
| | IngeniOs β-TCP (Zimmer Dental, CA, USA) |
| | ISIOS+ (Kasios, France) |
| | JAX (Smith and Nephew Orthopaedics, USA) |
| | Keramedic (Keramat, Spain) |
| | KeraOs (Keramat, Spain) |
| | Mega-TCP (Megagen Implant, Korea) |
| | microTCP (Conmed, USA) |
| | nanoXIM (Fluidinova, Portugal) |
| | Orthograft (DePuy Spine, USA) |
| | Ossaplast (Ossacur, Germany) |
| | Osferion (Olympus Terumo Biomaterials, Japan) |
| | Osfill (Olympus Terumo Biomaterials, Japan) |
| | OsproLife β-TCP (Lincotek Medical, Italy) |
| | OsSatura TCP (Integra Orthobiologics, CA, USA) |
| | Ossoconduct (SteinerBio, NV, USA) |
| | Osteoblast (Galimplant, Spain) |
| | Osteocera (Hannox, Taiwan) |
| | Osteopore TCP (SpiteCraft, IL, USA) |
| | OSTEOwelt (Biolot Medical, Turkey) |
| | Periophil β-TCP (Cytophil, WI, USA) |
| | Platon Pearl Bone (Platon, Japan) |
| | PolyBone (Kyungwon Medical, Korea) |
| | PORESORB-TCP (Lasak, Czech Republic) |
| | Powerbone (Medical Expo Bonegraft Biomaterials, Spain) |
| | PRINT3D Tricalcium Phosphate (Prodways, France) |
| | Repros (JRI Orthopaedics, UK) |
| | R.T.R. (Septodont, PA, USA) |
| | SigmaOs TCP (SigmaGraft, CA, USA) |
| | Socket Graft (SteinerBio, NV, USA) |
| | Sorbone (Meta Biomed, Korea) |
| | SUPERPORE (HOYA Technosurgical, Japan) |
| | Suprabone TCP (BMT Group, Turkey) |
| | Syncera (Oscotec, Korea) |
| | SynthoGraft (Bicon, MA, USA) |
| | Synthos (unknown producer) |
| | Syntricer (KARL STORZ Recon, Germany) |
| | TCP (Kasios, France) |
| | Terufill (Olympus Terumo Biomaterials, Japan) |
| | TKF-95 (Polystom, Russia) |

**Table 3.** *Cont*.

| Calcium Orthophosphate | Trade Name and Producer (When Available) |
|---|---|
| | TriCaFor (BioNova, Russia) |
| | Triha+ (Teknimed, France) |
| | TriOSS (Bioceramed, Portugal) |
| | Vitomatrix (Orthovita, PA, USA) |
| | Vitoss (Orthovita, PA, USA) |
| | without trade name (CaP Biomaterials, WI, USA) |
| | without trade name (Cam Bioceramics, Netherlands) |
| | without trade name (DinganTec, China) |
| | without trade name (Ensail Beijing, China) |
| | without trade name (Himed, NY, USA) |
| | without trade name (Shanghai Bio-lu Biomaterials, China) |
| | without trade name (Shanghai Rebone Biomaterials, China) |
| | without trade name (SigmaGraft, CA, USA) |
| | without trade name (Taihei Chemical Industrial, Japan) |
| | without trade name (Xpand Biotechnology, Netherlands) |
| β-TCP/CaSO$_4$ | Fortoss vital (Biocomposites, UK) |
| | Genex (Biocomposites, UK) |
| β-TCP/poly-lactic acid | Bilok (Biocomposites, UK) |
| | Duosorb (SBM, France) |
| | Matryx Interference Screws (Conmed, USA) |
| β-TCP/poly-lactic-*co*-glycolic acid | Evolvemer TCP30PLGA (Arctic Biomaterials, Finland) |
| β-TCP/polymer | AttraX putty (NuVasive, CA, USA) |
| | Therigraft (Therics, OH, USA) |
| β-TCP/bone marrow aspirate | Induce (Skeletal Kinetics, CA, USA) |
| β-TCP/collagen | Integra Mozaik (Integra Orthobiologics, CA, USA) |
| β-TCP/growth-factor | GEM 21S (Lynch Biologics, USA) |
| β-TCP/rhPDGF-BB solution | AUGMENT Bone Graft (Wright Medical Group, TN, USA) |
| | 4Bone BCH (MIS Implants, Israel) |
| | adboneBCP (Medbone Medical Devices, Portugal) |
| | Antartik (MedicalBiomat, France) |
| | ARCA BONE (ARCA-MEDICA, Switzerland) |
| | Artosal (aap Implantate, Germany) |
| | BABI-HATCP (Berkeley Advanced Biomaterials, CA, USA) |
| | Bicera (Hannox, Taiwan) |
| | BCP BiCalPhos (Medtronic, MN, USA) |
| | BIO-C (Cowellmedi, Korea) |
| | BioActys (Graftys, France) |
| | BioGraft (IFGL BIO CERAMICS, India) |
| | Biosel (Depuy Bioland, France) |
| | BonaGraft (Biotech One, Taiwan) |
| | Boncel-Os (BioAlpha, Korea) |

**Table 3.** *Cont.*

| Calcium Orthophosphate | Trade Name and Producer (When Available) |
| --- | --- |
| | Bone Plus BCP (Megagen Implant, Korea) |
| | Bone Plus BCP Eagle Eye (Megagen Implant, Korea) |
| | BoneMedik-DM (Meta Biomed, Korea) |
| | BoneSave (Stryker Orthopaedics, NJ, USA) |
| | BoneSigma BCP (SigmaGraft, CA, USA) |
| | BONITmatrix (DOT, Germany) |
| | Calcicoat (Zimmer, IN, USA) |
| | Calciresorb (Ceraver, France) |
| | Calc-i-oss crystal (Degradable Solutions, Switzerland) |
| | CellCeram (Scaffdex, Finland) |
| | Ceraform (Teknimed, France) |
| | Ceratite (NGK Spark Plug, Japan) |
| | Cross.Bone (Biotech Dental, France) |
| | CuriOs (Progentix Orthobiology BV, Netherlands) |
| | DM-Bone (Meta Biomed, Korea) |
| | Eclipse (Citagenix, QC, Canada) |
| | Eurocer (FH Orthopedics, France) |
| | Frabone (Inobone, Korea) |
| | Genesis-BCP (DIO, Korea) |
| | GenPhos HA TCP (Baumer, Brazil) |
| | Graftys BCP (Graftys, France) |
| | Hatric (Arthrex, Naples, FL, USA) |
| | Hydroxyapol (Polystom, Russia) |
| | Kainos (Signus, Germany) |
| | MagnetOs (Kuros Biosciences, Switzerland) |
| | MasterGraft (Medtronic Sofamor Danek, TN, USA) |
| | Maxresorb (botiss, Germany) |
| | MBCP (Biomatlante, France) |
| | MimetikOss (Mimetis Biomaterials, Spain) |
| | Neobone (Bioceramed, Portugal) |
| | New Bone (GENOSS, Korea) |
| | NT-BCP (OssGen, Korea) |
| | NT-Ceram (Meta Biomed, Korea) |
| | OdonCer (Teknimed, France) |
| | OpteMX (Exactech, FL, USA) |
| | OrthoCer HA TCP (Baumer, Brazil) |
| | OsproLife HA-βTCP (Lincotek Medical, Italy) |
| | OsSatura BCP (Integra Orthobiologics, CA, USA) |
| | ossceram nano (bredent medical, Germany) |
| | OSSEOPLUS (JHS Biomateriais, Brazil) |
| | Osspol (Genewel, Korea) |

**Table 3.** *Cont.*

| Calcium Orthophosphate | Trade Name and Producer (When Available) |
|---|---|
| | OsteoFlux (VIVOS-Dental, Switzerland) |
| | Osteon (GENOSS, Korea) |
| | Osteosynt (Einco, Brazil) |
| | Ostilit (Stryker Orthopaedics, NJ, USA) |
| | Ovis Bone BCP (DENTIS, Korea) |
| | Periophil biphasic (Cytophil, WI, USA) |
| | Q-OSS+ (Osstem Implant, Korea) |
| | ReproBone (Ceramisys, UK) |
| | R.T.R.+ (Septodont, PA, USA) |
| | SBS (Expanscience, France) |
| | Scaffdex (Scaffdex Oy, Finland) |
| | SigmaOs BCP (SigmaGraft, CA, USA) |
| | SinboneHT (Purzer Pharmaceutical, Taiwan) |
| | SkeliGraft (Osteotec, UK) |
| | Straumann BoneCeramic (Straumann, Switzerland) |
| | SYMBIOS Biphasic Bone Graft Material (DENTSPLY Implants, Sweden) |
| | SynMax (BioHorizons, Spain) |
| | Synergy (unknown producer) |
| | TCH (Kasios, France) |
| | Topgen-S (Toplan, Korea) |
| | Tribone (Stryker, Europe) |
| | Triosite (Zimmer, IN, USA) |
| | without trade name (AlgOss Biotechnologies, Austria) |
| | without trade name (Cam Bioceramics, Netherlands) |
| | without trade name (CaP Biomaterials, WI, USA) |
| | without trade name (Himed, NY, USA) |
| | without trade name (MedicalGroup, France) |
| | without trade name (SigmaGraft, CA, USA) |
| | without trade name (Xpand Biotechnology, Netherlands) |
| BCP (HA + α-TCP) | Skelite (Millennium Biologix, ON, Canada) |
| | Allograft (Zimmer, IN, USA) |
| | collacone max (botiss, Germany) |
| | Collagraft (Zimmer, IN, USA) |
| | Cross.Bone Matrix (Biotech Dental, France) |
| BCP (HA + β-TCP)/collagen | Indost (Polystom, Russia) |
| | MasterGraft (Medtronic Sofamor Danek, TN, USA) |
| | MATRI BONE (Biom'Up, France) |
| | Osteon III collagen (GENOSS, Korea) |
| | SynergOss (Nobil Bio Ricerche, Italy) |
| | without trade name (MedicalGroup, France) |

**Table 3.** *Cont*.

| Calcium Orthophosphate | Trade Name and Producer (When Available) |
|---|---|
| BCP (HA + β-TCP)/hydrogel | 4MATRIX+ (MIS Implants, Israel) |
| | Eclipse (Citagenix, QC, Canada) |
| BCP (HA + β-TCP)/polymer | In'Oss (Biomatlante, France) |
| | Hydros (Biomatlante, France) |
| | Osteocaf (Texas Innovative Medical Devices, TX, USA) |
| | Osteotwin (Biomatlante, France) |
| BCP (HA + TTCP) | OsproLife HA-TTCP (Lincotek Medical, Italy) |
| BCP (HA + β-TCP)/chitosan | k-IBS (Bioceramed, Portugal) |
| BCP (HA + β-TCP)/fibrin | TricOS (Baxter BioScience, France) |
| BCP (HA + β-TCP)/silicon | FlexHA (Xomed, FL, USA) |
| Bioglass + α-TCP + β-TCP + HA + polymers | OsteoFlo NanoPutty (SurGenTec, FL, USA) |
| FA | without trade name (CaP Biomaterials, WI, USA) |
| FA + BCP (HA + β-TCP) | FtAP (Polystom, Russia) |
| DCPA | without trade name (Himed, NY, USA) |
| | without trade name (Shanghai Rebone Biomaterials, China) |
| DCPA + MgHPO$_4$·3H$_2$O + SiO$_2$ + carboxymethyl cellulose | Novogro (OsteoNovus, OH, USA) |
| DCPD | without trade name (Himed, NY, USA) |
| DCPD/collagen | CopiOs Bone Void Filler (Zimmer, IN, USA) |
| DCPD + β-TCP/CaSO$_4$ | PRO-DENSE (Wright Medical Group, TN, USA) |
| DCPD + β-TCP/CaSO$_4$ + collagen | PRO-STIM (Wright Medical Group, TN, USA) |
| ACP | CAPTAL ACP (Plasma Biotal, UK) |
| | without trade name (Himed, NY, USA) |
| OCP | Bontree (HudensBio, Korea) |
| | OctoFor (BioNova, Russia) |
| | without trade name (Himed, NY, USA) |
| OCP/fibrin | FibroFor (BioNova, Russia) |
| OCP/collagen | Bonarc (Toyobo, Japan) |
| TTCP | without trade name (Ensail Beijing, China) |
| | without trade name (Himed, NY, USA) |
| | without trade name (Shanghai Rebone Biomaterials, China) |
| | without trade name (Taihei Chemical Industrial, Japan) |
| Undisclosed CaPO$_4$ | Arex Bone (Osteotec, UK) |
| | Inno-CaP (Cowellmedi, Korea) |
| Undisclosed CaPO$_4$ + biologics | i-FACTOR (Cerapedics, CO, USA) |
| MCPM | Phosfeed MCP (OCP group, Morocco) |
| MCPM + DCPD | Phosfeed MDCP (OCP group, Morocco) |

In spite of the aforementioned serious mechanical limitations (see *Section 4.1. Mechanical Properties*), bioceramics of CaPO$_4$ are available in various physical forms: powders, particles, granules (or granulates), dense blocks, porous scaffolds, self-setting formulations, implant coatings, and composite components of different origin (natural, biological, or

synthetic), often with specific shapes, such as implants, prostheses, or prosthetic devices. In addition, CaPO$_4$ are also applied as nonhardening injectable formulations [721–726] and pastes [726–730]. Generally, they consist of a mixture of CaPO$_4$ powders or granules and a "glue", which can be a highly viscous hydrogel. More to the point, custom-designed shapes such as wedges for tibial opening osteotomy, cones for spine and knee, and inserts for vertebral cage fusion are also available [546]. Various trademarks of the commercially available types of CaPO$_4$-based bioceramics and biomaterials are summarized in Table 3, while their surgical applications are schematically shown in Figure 15 [731]. A long list of both trademarks and producers clearly demonstrates that CaPO$_4$ bioceramics are easy to make and not very difficult to register for biomedical applications. There is an ISO standard for CaPO$_4$-based bone substitutes [732].

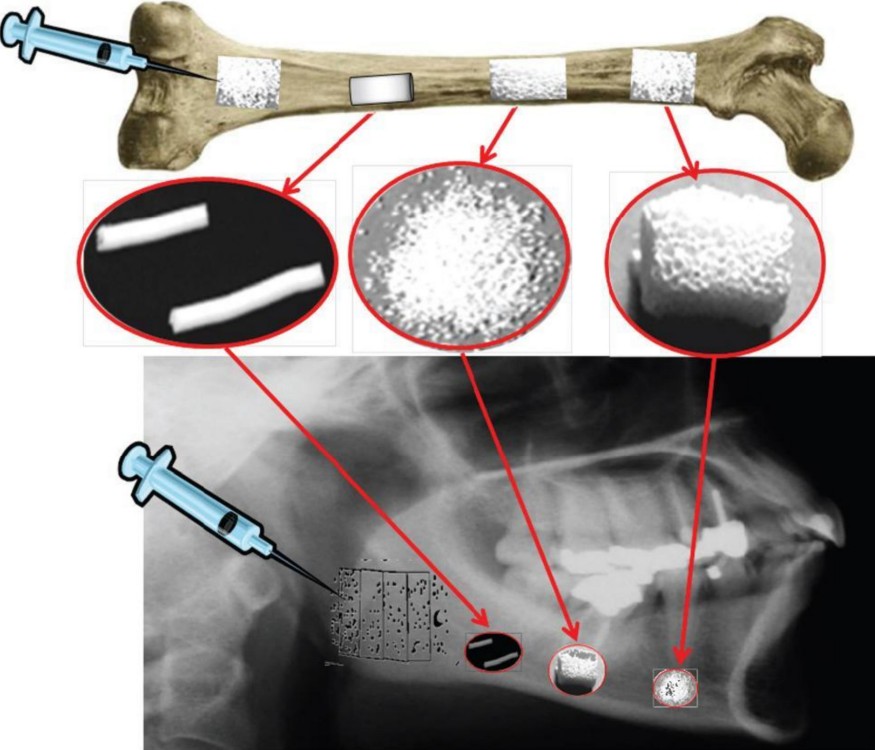

**Figure 15.** Different types of biomedical applications of CaPO$_4$ bioceramics. Reprinted from Ref. [731] with permission.

One should note that among the existing CaPO$_4$ (Table 1), only certain compounds are useful for biomedical applications, because those having a Ca/P ionic ratio less than 1 are not suitable for implantation due to their high solubility and acidity. Furthermore, due to its basicity, TTCP alone cannot be suitable either. Nevertheless, researchers try [141]. In addition, to simplify biomedical applications, these "of little use" CaPO$_4$ can be successfully combined with either other types of CaPO$_4$ or other chemicals.

### 6.1. Self-Setting (Self-Hardening) Formulations

The need for bioceramics for minimal invasive surgery has induced the concept of self-setting (or self-hardening) formulations consisting of CaPO$_4$ only to be applied as injectable and/or moldable bone substitutes [102,103,124,507,733]. After hardening, they form bulk CaPO$_4$ bioceramics. In addition, there are reinforced formulations that, in a certain sense, might be defined as CaPO$_4$ concretes [102]. Furthermore, self-setting formulations able to produce porous bulk CaPO$_4$ bioceramics are also available [450,451,458–461,507,520,733–736].

All types of the self-setting CaPO$_4$ formulations belong to low-temperature bioceramics. They are divided into two major groups. The first one is a dry mixture of two different types of CaPO$_4$ (a basic one and an acidic one), in which, after being wetted, the setting

reaction occurs according to an acid–base reaction. The second group contains only one $CaPO_4$, such as ACP with Ca/P molar ratio within 1.50–1.67 or α-TCP: both of them form CDHA upon contact with an aqueous solution [102,124]. Chemically, setting (= hardening, curing) is due to the succession of dissolution and precipitation reactions. Mechanically, it results from crystal entanglement and intergrowth (Figure 16) [737]. By influencing dimensions of forming $CaPO_4$ crystals, it is possible to influence the mechanical properties of the hardened bulk bioceramics [738]. Sometimes, the self-set formulations are sintered to prepare high-temperature $CaPO_4$ bioceramics [739]. Despite a large number of initial compositions, all types of self-setting $CaPO_4$ formulations can form three products only: CDHA, DCPD, and, rarely, DCPA [102,103,124,507,733]. Special reviews on the topic are available in [102,103,739], where interested readers are referred for further details.

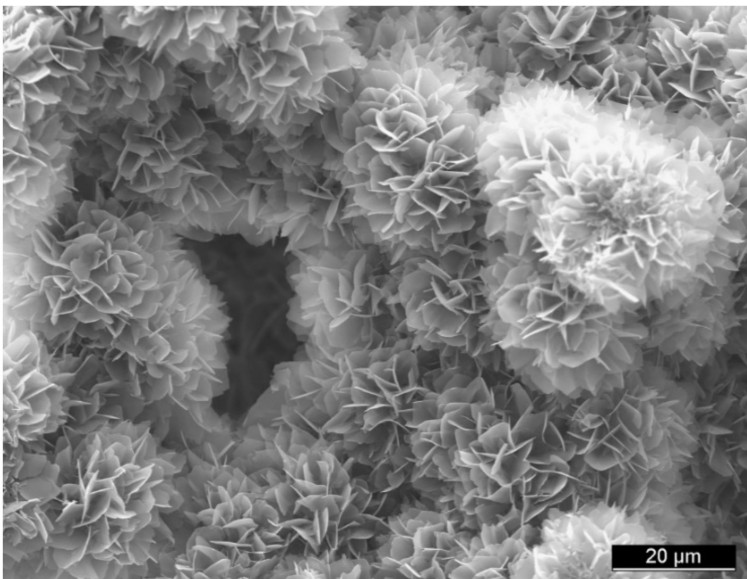

**Figure 16.** A typical microstructure of a $CaPO_4$ cement after hardening. The mechanical stability is provided by the physical entanglement of crystals. Reprinted from Ref. [737] with permission.

*6.2. $CaPO_4$ Deposits (Coatings, Films, and Layers)*

For many years, the clinical application of $CaPO_4$-based bioceramics has been largely limited to non-load-bearing parts of the skeleton due to their inferior mechanical properties. Therefore, materials with better mechanical properties appear to be necessary. For example, metallic implants are encountered in endoprostheses (total hip joint replacements) and artificial teeth sockets. As metals do not undergo bone bonding, i.e., they do not form a mechanically stable link between the implant and bone tissue, methods have been sought to improve contacts at the interface. One major method is to coat metals with $CaPO_4$, which enables bonding ability between the metal and the bone [180,190,397,740–742].

A number of factors influence the properties of $CaPO_4$ deposits (coatings, films, and layers). They include thickness (this will influence coating adhesion and fixation—the agreed optimum now seems to be within 50–100 μm), crystallinity (this affects the dissolution and biological behavior), phase and chemical purity, porosity, and adhesion. The coated implants combine the surface biocompatibility and bioactivity of $CaPO_4$ with the core strength of strong substrates (Figure 17). Moreover, $CaPO_4$ deposits decrease a release of potentially hazardous chemicals from the core implant and shield the substrate surface from environmental attack. In the case of porous implants, the $CaPO_4$-coated surface enhances bone ingrowth into the pores [331]. The clinical results for $CaPO_4$-deposited implants reveal that they have much longer lifetimes after implantation than uncoated devices and they are found to be particularly beneficial for younger patients. Further details on this topic are available in the special reviews [740–742].

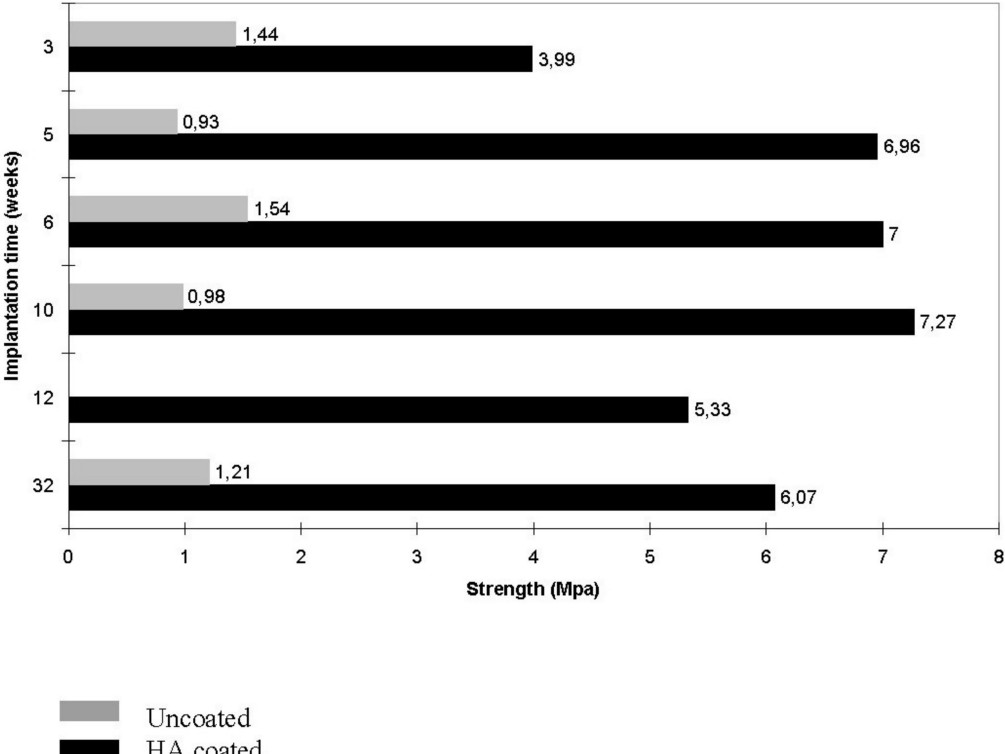

**Figure 17.** Shows how a plasma-sprayed HA coating on a porous titanium (dark bars), dependent on the implantation time, will improve the interfacial bond strength compared to uncoated porous titanium (light bars). Reprinted from Ref. [50] with permission.

*6.3. Functionally Graded Bioceramics*

In general, functionally gradient materials (FGMs) are defined as materials having either compositional or structural gradient from their surface to the interior. The idea of FGMs allows one device to possess two different properties. One of the most important combinations for the biomedical field is that of mechanical strength and biocompatibility. Namely, only surface properties govern a biocompatibility of the entire device. In contrast, the strongest material determines the mechanical strength of the entire device. Although this subject belongs to the previous section on coatings, films, and layers, in a certain sense, all types of implants covered by $CaPO_4$ might be also considered as FGMs.

Within the scope of this review, functionally graded bioceramics consisting of $CaPO_4$ are considered and discussed only. Such formulations have been developed [74,491,494, 550,743–753]. For example, dense sintered bodies with gradual compositional changes from $\alpha$-TCP to HA were prepared by sintering diamond-coated HA compacts at 1280 °C under a reduced pressure, followed by heating under atmospheric conditions [743]. The content of $\alpha$-TCP gradually decreased, while the content of HA increased with increasing depth from the surface. This functionally gradient bioceramic consisting of HA core and $\alpha$-TCP surface showed potential value as a bone-substituting biomaterial [743]. Two types of functionally gradient FA/$\beta$-TCP biocomposites were prepared in another study [744]. As shown in Figure 18, one of the graded biocomposites was in the shape of a disk and contained four different layers of about 1 mm thick. The other graded biocomposite was also in the shape of a disk but contained two sets of the four layers, each layer being 0.5 mm thick controlled by using a certain amount of the mixed powders. The final FA/$\beta$-TCP graded structures were formed at 100 MPa and sintered at 1300 °C for 2 h [744]. The same approach was used in yet another study, but HA was used instead of FA and CDHA was used instead of $\beta$-TCP [752]. $CaPO_4$ coatings with graded crystallinity were prepared as well [748].

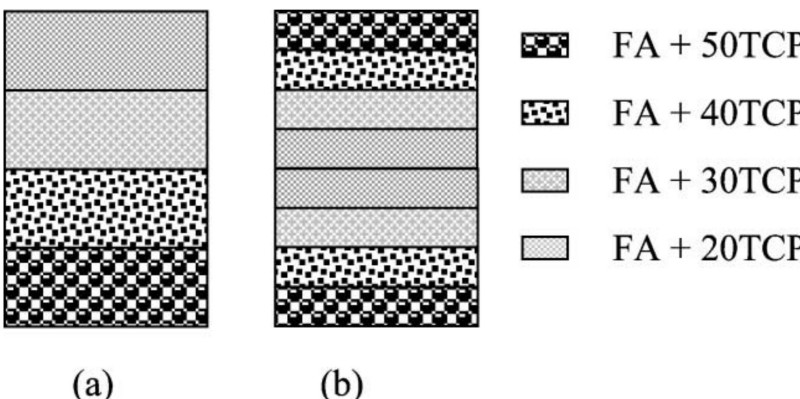

**Figure 18.** A schematic diagram showing the arrangement of the FA/β-TCP biocomposite layers. (**a**) A nonsymmetric functionally gradient material (FGM); (**b**) symmetric FGM. Reprinted from Ref. [744] with permission.

In addition, it is well known that a bone cross-section from cancellous to cortical bone is nonuniform in porosity and pore dimensions. Thus, in various attempts to mimic the porous structure of bones, $CaPO_4$ bioceramics with graded porosity have been fabricated [74,432,478,491,494,550,743–746]. For example, graded porous $CaPO_4$ bioceramics can be produced by means of tape casting and lamination (Figure 19, top). Other manufacturing techniques, such as a compression molding process (Figure 19, bottom) followed by impregnation and firing, are known as well [432]. In the first method, an HA slurry was mixed with a pore former. The mixed slurry was then cast into a tape. Using the same method, different tapes with different pore former sizes were prepared individually. The different tape layers were then laminated together. Firing was then performed to remove the pore formers and sinter the HA particle compacts, resulting in graded porous bioceramics [746]. This method was also used to prepare graded porous HA with a dense part (core or layer) in order to improve the mechanical strength, as dense ceramics are much stronger than porous ceramics. However, as in the pressure infiltration of mixed particles, this multiple tape casting also has the problem of poor connectivity of pores, although the pore size and the porosity are relatively easy to control. Furthermore, the lamination step also introduces additional discontinuity of the porosity on the interfaces between the stacked layers.

Since diverse biomedical applications require different configurations and shapes, the graded (or gradient) porous bioceramics can be grouped according to both the overall shape and the structural configuration [432]. The basic shapes include rectangular blocks and cylinders (or disks). For the cylindrical shape, there are configurations of dense core–porous layer, less porous core–more porous layer, dense layer–porous core, and less porous layer–more porous core. For the rectangular shape, in the gradient direction, i.e., the direction with varying porosity, pore size, or composition, there are configurations of porous top–dense bottom (same as porous bottom–dense top), porous top–dense center–porous bottom, dense top–porous center–dense bottom, etc. Concerning biomedical applications, a dense core–porous layer structure is suitable for implants of a high mechanical strength and with bone ingrowth for stabilization, whereas a less porous layer–more porous core configuration can be used for drug delivery systems. Furthermore, a porous top –dense bottom structure can be shaped into implants of articulate surfaces for wear resistance and with porous ends for bone ingrowth fixation, while a dense top–porous center–dense bottom arrangement mimics the structure of head skull. Further details on bioceramics with graded porosity can be found in the literature [432].

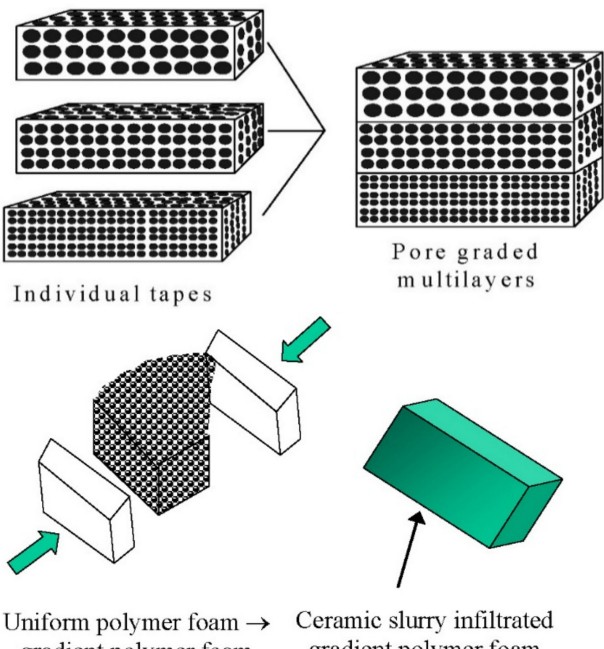

**Figure 19.** Schematic illustrations of fabrication of pore-graded bioceramics: top—lamination of individual tapes, manufactured by tape casting; bottom—a compression molding process. Reprinted from Ref. [432] with permission.

## 7. CaPO$_4$ Bioceramics in Tissue Engineering

### 7.1. Tissue Engineering

Tissue/organ repair has been the ultimate goal of surgery from ancient times to nowadays [56,57]. The repair has traditionally taken two major forms: tissue grafting followed by organ transplantation, and alloplastic or synthetic material replacement. Both approaches, however, have limitations. Grafting requires second surgical sites with associated morbidity and is restricted by limited amounts of material, especially for organ replacement. Synthetic materials often integrate poorly with host tissue and fail over time due to wear and fatigue or adverse body response [754]. In addition, all modern artificial orthopedic implants lack three of the most critical abilities of living tissues: (i) self-repairing; (ii) maintaining of blood supply; (iii) self-modifying their structure and properties in response to external aspects such as a mechanical load [755]. It is needless to mention that bones not only possess all of these properties but, in addition, they are self-generating, hierarchical, multifunctional, nonlinear, composite, and biodegradable; therefore, the ideal artificial bone grafts must possess similar properties [62].

The last decades have seen a surge in creative ideas and technologies developed to tackle the problem of repairing or replacing diseased and damaged tissues, leading to the emergence of a new field in healthcare technology now referred to as *tissue engineering*, which might be defined as "the creation of new tissue for the therapeutic reconstruction of the human body, by the deliberate and controlled stimulation of selected target cells through a systematic combination of molecular and mechanical signals" [756]. Briefly, this is an interdisciplinary field that exploits a combination of living cells, engineering materials, and suitable biochemical factors (Figure 20) in a variety of ways to improve, replace, restore, maintain, or enhance living tissues and whole organs [757–759]. However, since two of three major components (namely, cells and biochemical factors) of the tissue engineering subject appear to be far beyond the scope of this review, the topic of bone tissue engineering that aims to mimic the in vivo bone regeneration processes in a laboratory environment is narrowed down to the engineering materials prepared from CaPO$_4$ bioceramics only.

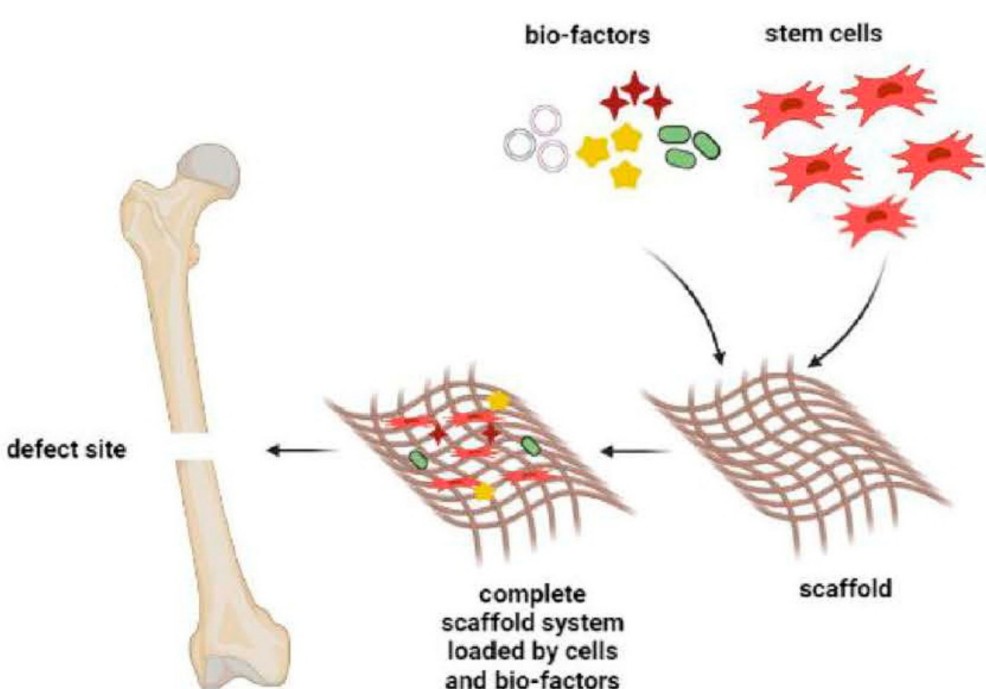

**Figure 20.** A tissue engineering approach for developing an advanced bone scaffold. Reprinted from Ref. [759] with permission.

Regeneration, rather than a repair, is the central goal of any tissue engineering strategy; therefore, it aims to create tissues and organs de novo [758]. This field of science started more than two decades ago [760,761], and the famous publication by Langer and Vacanti [762] has greatly contributed to the promotion of tissue engineering research worldwide. The field of tissue engineering, particularly when applied to bone substitutes where tissues often function in a mechanically demanding environment [763–765], requires a collaboration of excellence in cell and molecular biology, biochemistry, material sciences, bioengineering, and clinical research [766]. For success, it is necessary that researchers with expertise in one area have an appreciation of the knowledge and challenges of the other areas. However, since the technical, regulatory, and commercial challenges might be substantial, the introduction of new products is likely to be slow [758].

Nowadays, tissue engineering is at full research potential due to the following key advantages: (i) the solutions it provides are long-term, much safer than other options, and cost-effective as well; (ii) the need for donor tissue is minimal, which eliminates the immunosuppression problems; (iii) the presence of residual foreign material is eliminated as well [767,768].

### 7.2. Scaffolds and Their Properties

It would be very convenient for both patients and physicians if devastated tissues or organs of patients could be regenerated by simple cell injections to the target sites, but such cases are rare. The majority of large-sized tissues and organs with distinct 3D form require a support for their formation from cells. The support is called a scaffold, template, and/or artificial extracellular matrix [127,128,534,760,763–772]. The major function of scaffolds is similar to that of the natural extracellular matrix that assists proliferation, differentiation, and biosynthesis of cells. In addition, scaffolds placed at the regeneration sites will prevent disturbing cells from invasion into the sites of action [771,772]. The role of scaffolds was perfectly described by a Spanish classical guitarist Andrés Segovia (1893–1987): "When one puts up a building one makes an elaborate scaffold to get everything into its proper place. But when one takes the scaffold down, the building must stand by itself with no trace of the means by which it was erected. That is how a musician should work". However, for the future of tissue engineering, the term "template" might become more suitable because,

according to David F. Williams, the term scaffold "conveys an old fashioned meaning of an inert external structure that is temporarily used to assist in the construction or repair of inanimate objects such as buildings, taking no part in the characteristics of the finished product." [773] (p. 1129).

Therefore, the idea behind tissue engineering is to create or engineer autografts by either expanding autologous cells in vitro guided by a scaffold or implanting an acellular template in vivo and allowing the patient's cells to repair the tissue guided by the scaffold. The first phase is the in vitro formation of a tissue constructed by placing the chosen cells and scaffolds in a metabolically and mechanically supportive environment with growth media (in a bioreactor), in which the cells proliferate and elaborate extracellular matrix. It is expected that cells infiltrate into the porous matrix and consequently proliferate and differentiate therein [774,775]. In the second phase, the construct is implanted in the appropriate anatomic location, where remodeling in vivo is intended to recapitulate the normal functional architecture of an organ or a tissue [776,777]. The key processes occurring during both in vitro and in vivo phases of the tissue formation and maturation are (1) cell proliferation, sorting, and differentiation, (2) extracellular matrix production and organization, (3) biodegradation of the scaffold, and (4) remodeling and potentially growth of the tissue [778].

To achieve the goal of tissue reconstruction, the scaffolds (templates) must meet a number of the specific requirements [127,128,769–773]. Five features of the scaffold's architecture appear to influence the biological response: (1) a macroscopic shape, (2) a porous network, (3) pore dimensions and geometry, (4) surface microtopography, and (5) micro-, submicro-, and nanoporosities. In addition, scaffolds should be biodegradable. Among them, a reasonable surface roughness is necessary to facilitate cell seeding and fixation [619,779–784]. A high porosity and the adequate pore dimensions (Tables 2 and 4) are very important to allow cell migration and vascularization, as well as a diffusion of nutrients [437,758]. A French architect, Robert le Ricolais (1894–1977), stated: "The art of structure is where to put the holes". Therefore, to enable proper tissue ingrowth, vascularization, and nutrient delivery, scaffolds should have a highly interconnected porous network, formed by a combination of macro- and micropores, in which more than ~60% of the pores should have a size ranging from ~150 to ~400 µm and at least ~20% should be smaller than ~20 µm [437,442,443,448,529,530,536,538,544,563–570,572,754,785–791]. Furthermore, a sufficient mechanical strength and stiffness are mandatory to oppose contraction forces and later for the remodeling of damaged tissues [792,793]. In addition, scaffolds must be manufactured from the materials with controlled biodegradability and/or bioresorbability, such as $CaPO_4$, so that a new bone will eventually replace the scaffold [763,786,794]. Furthermore, the degradation byproducts of scaffolds must be noncytotoxic. More to the point, the resorption rate has to coincide as much as possible with the rate of bone formation (i.e., between a few months and about 2 years) [795]. This means that while cells are fabricating their own natural matrix structure around themselves, the scaffold is able to provide a structural integrity within the body, and eventually it will break down, leaving the newly formed tissue that will take over the mechanical load. However, one should bear in mind that the scaffold's architecture changes with the degradation process, and the degradation byproducts affect the biological response. In addition, scaffolds should be easily fabricated into a variety of shapes and sizes [796] and be malleable to fit irregularly shaped defects, while the fabrication processes should be effortlessly scalable for mass production. In many cases, ease of processability, as well as easiness of conformation and injectability, which self-setting $CaPO_4$ formulations possess (see *Section 6.1. Self-setting (Self-hardening) Formulations*), can determine the choice of a certain biomaterial. Finally, sterilization with no loss of properties is a crucial step in scaffold production at both a laboratory and an industrial level [763–765]. Thus, each scaffold (template) should fulfill many functions before, during, and after implantation.

**Table 4.** A hierarchical pore size distribution that an ideal scaffold should exhibit [797].

| Pore Sizes of a 3D Scaffold | A Biochemical Effect or Function |
|---|---|
| <1 μm | Interaction with proteins |
| | Responsible for bioactivity |
| 1–20 μm | Type of cells attracted |
| | Cellular development |
| | Orientation and directionality of cellular ingrowth |
| 100–1000 μm | Cellular growth |
| | Bone ingrowth |
| | Predominant function in the mechanical strength |
| >1000 μm | Implant functionality |
| | Implant shape |
| | Implant esthetics |

Many fabrication techniques are available to produce porous $CaPO_4$ scaffolds (Table 2) with varying architectural features (see the aforementioned Section 3.3 and 4.4). In order to achieve the desired properties with the minimum expenses, the production process should be optimized [798]. The main goal is to develop a high potential synthetic bone substitute (so-called "smart scaffold") which will not only promote osteoconduction, i.e., bone growth on a surface, but also osteopromotion, i.e., the ability to enhance osteoinduction [799]. In the case of $CaPO_4$, a smart scaffold represents a biphasic (HA/β–TCP ratio of 20/80) formulation with a total porosity of ~73%, constituted of macropores (>100 μm), mesopores (10–100 μm), and a high content (~40%) of micropores (<10 μm) with the crystal dimensions within <0.5 to 1 μm and the specific surface area ~6m$^2$/g [800]. With the advent of $CaPO_4$ in tissue engineering, the search is on for the ultimate option consisting of a synthetic smart scaffold impregnated with cells and growth factors. Figure 21 schematically depicts a possible fabrication process of such item that, afterwards, will be implanted into a living organism to induce bone regeneration [47].

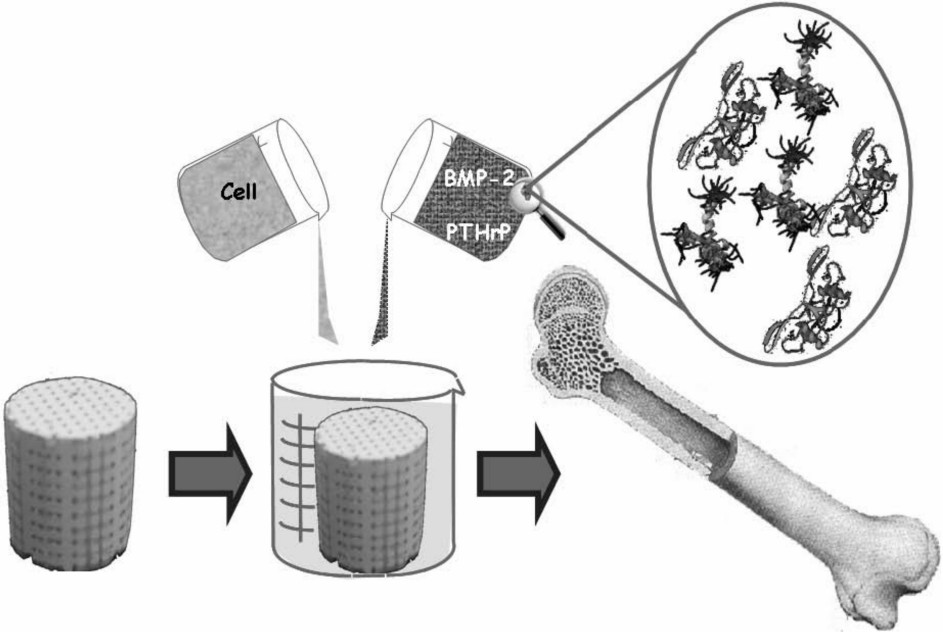

**Figure 21.** A schematic view of a third-generation biomaterial, in which porous $CaPO_4$ bioceramic acts as a scaffold or a template for cells, growth factors, etc. Reprinted from Ref. [47] with permission.

To finalize this topic, one should note the fundamental unfeasibility to create the so-called "ideal scaffold" for bone grafting. Since bones of the human skeleton have very different dimensions, shapes, and structures depending on their functions and locations, synthetic bone grafts of various sizes, shapes, porosity, mechanical strength, composition, and resorbability appear to be necessary. Therefore, HA bioceramics of 0 to 15% porosity are used as both ilium and intervertebral spacers, where a high strength is required, HA bioceramics of 30 to 40% porosity are useful as spinous process spacers for laminoplasty, where both bone formation and middle strength are necessary, while HA bioceramics of 40% to 60% porosity are useful for the calvarias plate, where a fast bone formation is needed (Figure 22) [518]. Furthermore, defining the optimum parameters for artificial scaffolds is, in fact, an attempt to find a reasonable compromise between various conflicting functional requirements. Namely, an increased mechanical strength of bone substitutes requires solid and dense structures, while colonization of their surfaces by cells requires interconnected porosity. Additional details and arguments on this subject are well described elsewhere [801], in which the authors concluded that "there is enough evidence to postulate that ideal scaffold architecture does not exist." (p. 478).

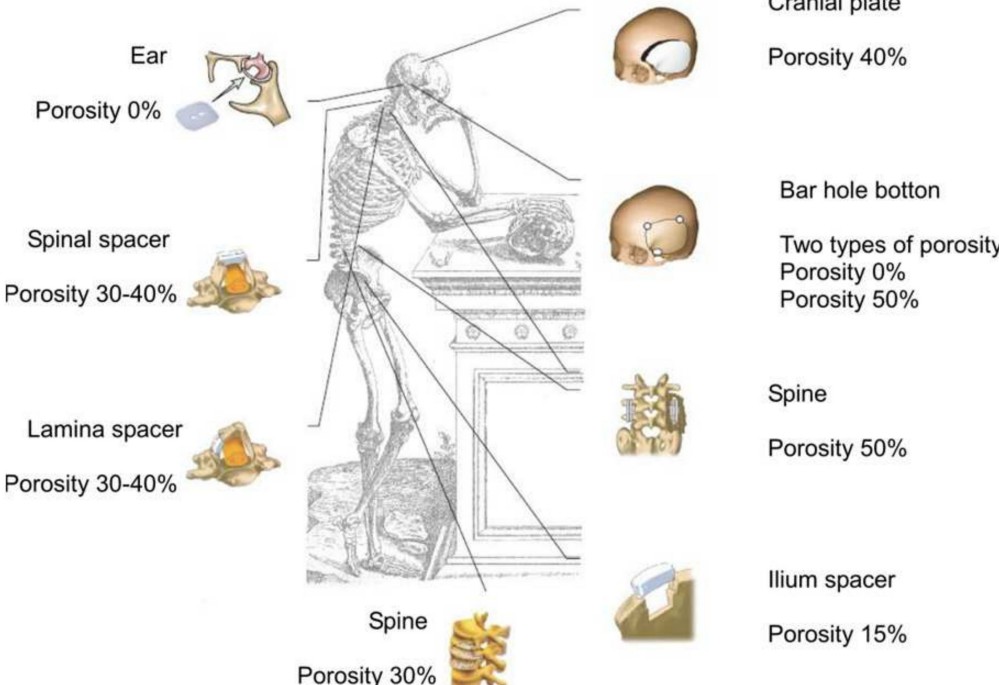

**Figure 22.** A schematic drawing presenting the potential usage of HA with various degrees of porosity. Reprinted from Ref. [518] with permission.

### 7.3. Bioceramic Scaffolds from CaPO$_4$

Philosophically, the increase in life expectancy requires biological solutions to all biomedical problems, including orthopedic ones, which were previously managed with mechanical solutions. Therefore, since the end of the 1990s, biomaterials research has focused on tissue regeneration instead of tissue replacement [802]. The alternatives include using hierarchical bioactive scaffolds to engineer in vitro living cellular constructs for transplantation or using bioresorbable bioactive particulates or porous networks to activate, in vivo, the mechanisms of tissue regeneration [803,804]. Thus, the aim of CaPO$_4$ is to prepare artificial porous bioceramic scaffolds able to provide the physical and chemical cues to guide cell seeding, differentiation, and assembly into 3D tissues of a newly formed bone. Particle sizes, shape, and surface roughness of the scaffolds are known to affect cellular adhesion, proliferation, and phenotype [619,779–784]. Additionally, the surface energy might play a role in attracting particular proteins to the bioceramic surface and,

in turn, this will affect the cells' affinity to the material. More to the point, cells are exceedingly sensitive to the chemical composition, and their bone-forming functions can be dependent on grain morphology of the scaffolds. For example, osteoblast functions were found to increase on nanodimensional fibers when compared to nanodimensional spheres because the former more closely approximated the shape of biological apatite in bones [805]. In addition, a significantly higher osteoblast proliferation on HA bioceramics sintered at 1200 °C, compared to that on HA bioceramics sintered at 800 and 1000 °C, was reported [806]. Furthermore, since ions of calcium and orthophosphate are known to regulate bone metabolism, $CaPO_4$ appears to be among the few bone graft substitute materials that can be considered as a drug. A schematic drawing of the key scaffold properties affecting a cascade of biological processes occurring after $CaPO_4$ implantation is shown in Figure 23 [807].

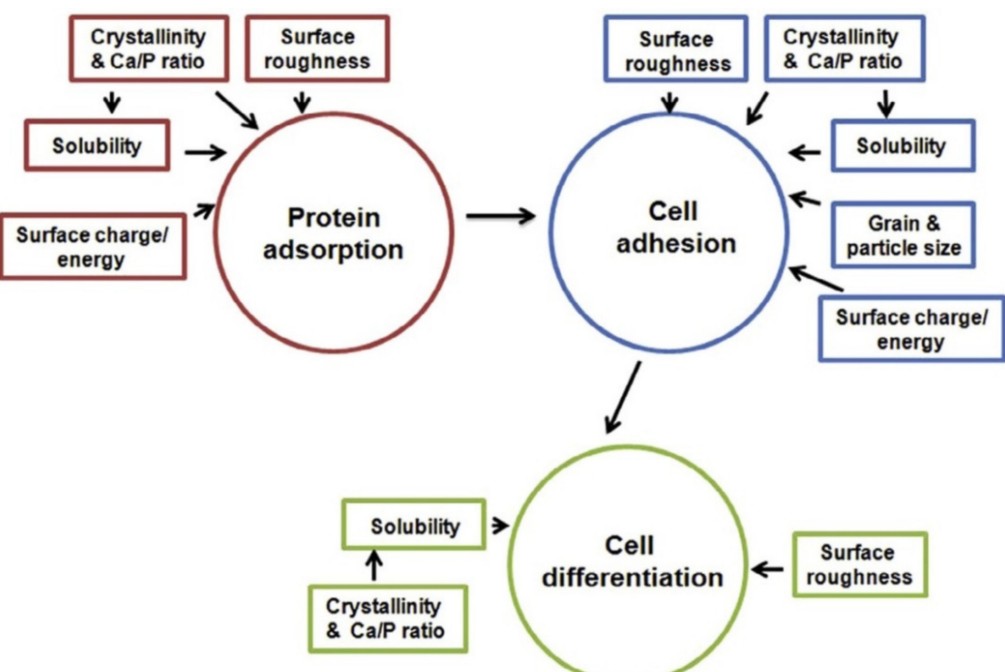

**Figure 23.** A schematic drawing of the key scaffold properties affecting a cascade of biological processes occurring after $CaPO_4$ implantation. Reprinted from Ref. [807] with permission.

Thus, to meet the tissue engineering requirements, much attention is devoted to further improvements of $CaPO_4$ bioceramics [808–810]. From the chemical point of view, the developments include synthesis of novel ion-substituted $CaPO_4$ [17–41]. For example, a recent systematic review and meta-analysis indicated a significant positive effect on new bone formation by supplementing $CaPO_4$-based bone substitutes with bioinorganics compared to those without dopants, especially when strontium, silicon, or magnesium were used [811]. A positive influence of $CaPO_4$ doping by carbonates was also noticed [812]. From the material point of view, the major research topics include nanodimensional and nanocrystalline structures [813–816], amorphous compounds [817,818], (bio)organic/$CaPO_4$ biocomposites and hybrid formulations [357,819,820], and biphasic, triphasic, and multiphasic formulations [79], as well as various types of structures, forms, and shapes. The latter comprise fibers, wires, whiskers, and filaments [135,231,821–834], macro-, micro-, and nanosized spheres, beads, and granules [833–846], tetrapods and pyramids [847], micro- and nanosized tubes [848–852], aerogels [853], "flowers" [854], porous 3D scaffolds [172] made of ACP [470,582,855], DCPD/DCPA [148,856–859], OCP [860,861], TCP [68,71,139,140,862–865], HA [147,447,448,487,513,514,798,852,866–870], TTCP [141] and biphasic formulations [245,474,491,531,836,843,865,871–877], structures with graded porosity [74,432,478,491,494,550,743–746], and hierarchically organized ones [877,878]. More

advanced combined techniques are also possible. For example, to increase the degree of os-sification and homogenization, the surface of porous CaPO$_4$ scaffolds might be modified by generation of CaPO$_4$ whiskers on their surface [879,880], followed by reinforcing with multi-ple layers of releasable nanodimensional CDHA particles [880]. Recently, research progress has been made in the deformable CaPO$_4$-based biomaterials with a high flexibility, softness, and/or elasticity based on nanodimensional ultralong HA wires [881,882]. Furthermore, an addition of defects through an intensive milling [883,884] or their removal by a thermal treatment [885] can be used to modify the chemical reactivity of CaPO$_4$. In addition, more attention should be paid to a crystallographically aligned CaPO$_4$ bioceramics [687–692,886].

In general, there are three principal therapeutic strategies for treating diseased or injured tissues in patients: (i) implantation of freshly isolated or cultured cells; (ii) implan-tation of tissues assembled in vitro from cells and scaffolds; (iii) in situ tissue regeneration. For cellular implantation, individual cells or small cellular aggregates from the patient or a donor are either injected into the damaged tissue directly or are combined with a degradable scaffold in vitro and then implanted. For tissue implantation, a complete 3D tissue is grown in vitro using patient or donor cells and a bioresorbable scaffold and then is implanted into the patients to replace diseased or damaged tissues. For in situ regeneration, a scaffold implanted directly into the injured tissue stimulates the body's own cells to pro-mote local tissue repair [324,757]. In any case, simply trapping cells at the particular point on a surface is not enough: the cells must be encouraged to differentiate, which is impossi-ble without the presence of suitable biochemical factors [887]. All previously mentioned points clearly indicate that, for the purposes of tissue engineering, CaPO$_4$ bioceramics play an auxiliary role; namely, they acts as a suitable material to manufacture the appropriate 3D templates, substrates, or scaffolds to be colonized by living cells before the successive implantation [799,800,888–890]. The in vitro evaluation of potential CaPO$_4$ scaffolds for tissue engineering is described elsewhere [891], while the data on the mechanical properties of CaPO$_4$ bioceramics for use in tissue engineering are also available [892–894]. The effect of an HA-based biomaterial on gene expression in osteoblast-like cells was reported as well [895]. To conclude this part, the excellent biocompatibility of CaPO$_4$ bioceramics, their possible osteoinductivity [544,566,597–605], and high affinity for drugs [54–57,896–900], proteins, and cells [897,901] make them very functional for the tissue engineering appli-cations [902]. The feasible production of scaffolds with tailored structures and properties opens up a spectacular future for CaPO$_4$ bioceramics [895–903].

### 7.4. A Clinical Experience

To date, there are just a few publications on clinical application of cell-seeded CaPO$_4$ bioceramics for bone tissue engineering of humans. Namely, Quarto et al. [904] were the first to report a treatment of large (4–7 cm) bone defects of the tibia, ulna, and humerus in three patients aged from 16 to 41 years old, where the conventional surgical therapies had failed. The authors implanted a custom-made unresorbable porous HA scaffold seeded with in vitro expanded autologous bone marrow stromal cells. In all three patients, radiographs and computed tomographic scans revealed abundant callus formation along the implants and good integration at the interfaces with the host bones by the second month after surgery [904]. In the same year, Vacanti et al. [905] reported the case of a man who had a traumatic avulsion of the distal phalanx of a thumb. The phalanx was replaced with a specially treated natural coral (porous HA; 500-pore ProOsteon (see Table 3)) implant that was previously seeded with in vitro expanded autologous periosteal cells. The procedure resulted in the functional restoration of a stable and biomechanically sound thumb of normal length, without the pain and complications that are usually associated with harvesting a bone graft.

Morishita et al. [906] treated a defect resulting from surgery of benign bone tumors in three patients using HA scaffolds seeded with in vitro expanded autologous bone marrow stromal cells after osteogenic differentiation of the cells. Two bone defects in a tibia and one defect in a femur were treated. Although ectopic implants in nude mice

were mentioned to show the osteogenicity of the cells, details such as the percentage of the implants containing bone and at what quantities were not reported. Furthermore, cell-seeded $CaPO_4$ scaffolds were found to be superior to autograft, allograft or cell-seeded allograft in terms of bone formation at ectopic implantation sites [907]. In addition, it has been hypothesized that dental follicle cells combined with β-TCP bioceramics might become a novel therapeutic strategy to restore periodontal defects [908]. In yet another study, the behavior of human periodontal ligament stem cells on an HA-coated genipin-chitosan scaffold in vitro was studied followed by evaluation on bone repair in vivo [909]. The study demonstrated the potential of this formulation for bone regeneration.

A research group from Holland evaluated vascularization in relation to bone formation potential of adipose-stem-cells-containing stromal vascular fractions of adipose tissues, seeded on two types of $CaPO_4$ carriers, within the human maxillary sinus floor elevation model, in a phase I study [910]. Autologous stromal vascular fractions were obtained from 10 patients and seeded on either β-TCP scaffolds with 60% porosity (n = 5) or BCP (HA + β-TCP) ones with 90% porosity (n = 5) and used for maxillary sinus floor elevations in a one-step surgical procedure. After 6 months, biopsies were obtained during dental implant placements and the quantifications of the number of blood vessels were performed using histomorphometric analysis and immunohistochemical stainings for blood vessel markers. Bone percentages seemed to correlate with blood vessel formation and were higher in the study versus control biopsies in the cranial area, in particular for β-TCP-treated patients. That study showed the safety, feasibility, and efficiency of using of adipose-stem-cells-seeded $CaPO_4$ scaffolds for human maxillary sinus floor elevations and indicated a proangiogenic effect of stromal vascular fraction [910]. A brief description of several other cases is available in the literature [911].

To finalize this section, one must mention that $CaPO_4$ bioceramics are also used in veterinary orthopedics for favoring animal bone healing in areas in which bony defects exist [912,913].

## 8. Non-Biomedical Applications of $CaPO_4$

Due to their strong adsorption ability, surface acidity or basicity, and ion exchange abilities, some types of $CaPO_4$ possess a catalytic activity [914–926]. As seen from the references, $CaPO_4$ are able to catalyze oxidation and reduction reactions, as well as formation of C–C bonds. Namely, the application in oxidation reactions mainly includes oxidation of alcohol and dehydrogenation of hydrocarbons, while the reduction reactions include hydrogenolysis and hydrogenation. The formation of C–C bonds mainly comprises Claisen–Schmidt and Knoevenagel condensation reactions, Michael addition reaction, as well as Friedel–Crafts, Heck, Diels–Alder, and aldol reactions [921].

In addition, due to the chemical similarity to the inorganic part of mammalian calcified tissues, $CaPO_4$ powders appear to be good solid carriers for chromatography of biological substances. Namely, high-value biological materials such as recombinant proteins, therapeutic antibodies, and nucleic acids are separated and purified [927–933]. Furthermore, some types of $CaPO_4$ are used as a component of various sensors [373,374,378,379,382,934–938]. Finally, $CaPO_4$ ceramics appear to be good adsorbents of fluorides [939]; however, since these subjects are almost irrelevant to bioceramics, they are not detailed further. Additional details and examples are available elsewhere [940].

## 9. Conclusions and Outlook

The available chronology of seeking suitable bioceramics for bone substitutes is as follows: since the 1950s, the first aim was to use bioinert bioceramics, which had no reaction with living tissues. They included inert and tolerant compounds, which were designed to withstand physiological stress without, however, stimulating any specific cellular responses. Later on, in the 1980s, the trend changed towards exactly the opposite: the idea was to implant bioceramics that reacted with the surrounding tissues by producing newly formed bone (a "responsive" bioceramic because it was able to elicit biological responses).

These two stages are referred to as the first and the second generations of bioceramics, respectively [941] and, currently, both of them are extensively commercialized. Thus, the majority of the marketable products listed in Table 3 belong to the first and the second generations of bone substitute biomaterials. However, the progress has continued and, in the current century, scientists are searching for the third generation of bioceramics [324], which will be able to "instruct" the physiological environment toward desired biological responses (i.e., bioceramics will be able to regenerate bone tissues by stimulating specific responses at the molecular level) [47,942]. Since each generation represents an evolution of the requirements and properties of the biomaterials involved, one should stress that these three generations should not be interpreted as the chronological, but instead as the conceptual ones. This means that at present, research and development is still devoted to biomaterials and bioceramics that, according to their properties, could be considered to be of the first or the second generations, because the second generation of bioceramics with added porosity is one of the initial approaches in developing the third generation of bioceramics [943]. Furthermore, there is another classification of the history of biomaterials introduced by Prof. James M. Anderson. According to Anderson, from 1950–1975 the researchers studied bioMATERIALS, from 1975–2000 they studied BIOMATERIALS, and since 2000 the time for BIOmaterials has been approaching [944]. Here, the capital letters emphasis the major direction of the research efforts in the complex subject of biomaterials. As bioceramics are biomaterials of the ceramic origin (see *Section 2. General Knowledge and Definitions*), Anderson's historical classification appears to be applicable to the bioceramics field as well.

The historical development of biomaterials informs that their widespread use experiences two major difficulties. The first is an incomplete understanding of the physical and chemical functioning of biomaterials and of the human response to these materials. Recent advances in material characterization and computer science, as well as in cell and molecular biology, are expected to play a significant role in the study of biomaterials. A second difficulty is that many biomaterials do not perform as desirably as we would like. This is not surprising, since many materials used in medicine were not designed for medical purposes. It needs to be mentioned here that biomaterials are expected to perform in our body's internal environment, which is very aggressive. For example, solution pH of body fluids in various tissues varies in the range from 1 to 9. During daily activities, bones are subjected to a stress of ~4 MPa, whereas the tendons and ligaments experience peak stresses in the range of 40–80 MPa. The mean load on a hip joint is up to three times the body weight (3000 N) and peak load during jumping can be as high as ~10 times the body weight. More importantly, these stresses are repetitive and fluctuating, depending on the activities, such as standing, sitting, jogging, stretching, and climbing. All of these require careful designing of biomaterials in terms of composition, shape, physical, and biocompatibility properties. Therefore, a significant challenge is the rational design of human biomaterials based on a systematic evaluation of desired biological, chemical, and engineering requirements [945].

Nevertheless, the field of biomaterials is in the midst of a revolutionary change in which the life sciences are becoming equal in importance to materials science and engineering as the foundation of the field. Simultaneously, advances in engineering (for example, nanotechnology) are greatly increasing the sophistication with which biomaterials are designed and have allowed fabrication of biomaterials with increasingly complex functions [797]. Specifically, during the last ~50 years, $CaPO_4$ bioceramics have become an integral and vital segment of our modern healthcare delivery system. In the modern fields of the third-generation bioceramics (Hench) or BIOceramics (Anderson), the full potential of $CaPO_4$ has only begun to be recognized. Namely, $CaPO_4$, which were intended as osteoconductive bioceramics in the past, represent materials to fabricate osteoinductive implants nowadays [544,566,597–605]. Some steps in this direction have been already made by fabricating scaffolds for bone tissue engineering through the design of controlled 3D-porous structures and increasing the biological activity through development of novel ion-substituted $CaPO_4$ bioceramics [546,946]. The future of biosynthetic bone implants will

point to better mimicking the autologous bone grafts. Therefore, the composition, structure, and molecular surface chemistry of various types of $CaPO_4$ will be tailored to match the specific biological and metabolic requirements of tissues or disease states [947,948]. This new generation of $CaPO_4$ bioceramics should enhance the quality of life of millions of people as they grow older.

However, in spite of the great progress, there is still a great potential for major advances to be made in the field of $CaPO_4$ bioceramics. This includes requirements for [949]:

- Improvement of the mechanical performance of existing types of bioceramics.
- Enhanced bioactivity in terms of gene activation.
- Improvement in the performance of biomedical coatings in terms of their mechanical stability and ability to deliver biological agents.
- Development of smart biomaterials capable of combining sensing with bioactivity.
- Development of improved biomimetic composites.

Furthermore, there is still need for a better understanding of the biological systems. For example, the bonding mechanism between the bone mineral and collagen remains unclear. It is also unclear whether a rapid repair that is elicited by the new generation of bioceramics is a result of the enhancement of mineralization, per se, or whether there is a more complex signaling process involving proteins in collagen. If we were able to understand the fundamentals of bone response to specific ions and the signals they activate, then we would be able to design better bioceramics for the future [949].

To finalize this review, it is completely obvious that the present status of research and development in the field of $CaPO_4$ bioceramics is still at the starting point for the solution of new problems at the confluence of materials science, biology, and medicine, concerned with the restoration of damaged functions in the human organisms. A large increase in active elderly people has dramatically raised the need for load-bearing bone graft substitutes, for example, for bone reconstruction during revision arthroplasty or for the reinforcement of osteoporotic bones. Strategies applied in the last four decades towards this goal have failed, so new strategies, possibly based on self-assembling and/or nanofabrication, will have to be proposed and developed [950]. Angiogenesis (the process and stimulation of new blood vessel formation via sprouting from existing blood vessels) is also very important to the success of hard tissue regeneration, and $CaPO_4$ seem to be useful for this purpose [951]. In addition, some $CaPO_4$-containing formulations were tested with soft tissues [952,953], mainly for various types of soft-tissue augmentations [721,954–956] and eyeball replacements [704–709], as well as for cancer diagnostics and therapy [957–959], wound healing [960], as components of various cosmetic formulations [961], and vaccine adjuvants [962]. Recently, a concept of black bioceramics for Ca- and Mg-silicates, as well as for HA and TCP, was introduced [963]. The black bioceramics were prepared through a partial thermal reduction of traditional white ceramics ($CaSiO_3$, $MgSiO_3$, TCP, and HA) by magnesium. Due to the presence of oxygen vacancies and structural defects, the black bioceramics were found to possess a photothermal functionality while maintaining their initial high bioactivity and regenerative capacity. These black bioceramics showed excellent photothermal antitumor effects for both skin and bone tumors. At the same time, they significantly improved bioactivity for skin/bone tissue repair both in vitro and in vivo [963]. Bioceramics prepared from other types of calcium phosphates, such as calcium pyrophosphate, are worth investigating as well [964]. Additive manufacturing techniques will be further developed [965]. A bioinformatics approach to study the role of $CaPO_4$ properties in bone regeneration might be promising as well [966].

In future, it should be feasible to design a new generation of gene-activating $CaPO_4$-based scaffolds tailored for specific patients and disease states. In addition, further developments of 3D-printing technologies [967] will allow designing of personalized bone grafts, which will provide an accurate control of the geometry. The design of the implant shape, based on X-ray computed tomography data, will ensure the perfect fit between the graft and the anatomical defect [968]. Personalized implants will be produced with tailored characteristics better adapted to the patient-specific bone tissue regions/defects that need to

be replaced/reinforced. To transfer these technologies to clinical practice, material science and tissue engineering need to be closely assisted by biomedical researchers in order to confer the safety risk assessment, as well as efficacy at high standards. In addition, the development of complex testing strategies will help to unveil the network of biological events elicited by $CaPO_4$-based bioceramics as bulks, coatings, or nanodimensional forms, which are essential to ensure a longer and safer implant life in orthopedic and dentistry applications [969]. Perhaps, sometime-bioactive stimuli will be used to activate genes in a preventative treatment to maintain the health of aging tissues. Currently, all the afore-mentioned seem hardly possible. However, we need to remember that only ~50 years ago, the concept of a material that would not be rejected by living tissues also seemed impossible [583].

**Funding:** This research received no external funding.

**Institutional Review Board Statement:** Not applicable.

**Informed Consent Statement:** Not applicable.

**Data Availability Statement:** Not applicable.

**Conflicts of Interest:** The author declares no conflict of interest.

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
