# Peer review of "Calcium Orthophosphate (CaPO4)-Based Bioceramics: Preparation, Properties, and Applications"

_coatings, doi:10.3390/coatings12101380_

Round 1

Reviewer 1 Report

1.       Calcium orthophosphate is also involved in wound healing?

2.       Table 3 is too long. Divide them into 2 or more parts.

Author Response

Dear unknown reviewer!

First, thank you very much for spending your valuable time, kind efforts and suggestions to make my manuscript even better. Second, please, find below the point-to-point responses.

Yours sincerely,

Sergey V. Dorozhkin

Reviewer 1

Comments and Suggestions for Authors

  1. Calcium orthophosphate is also involved in wound healing? – Yes, but either as powders or a component of viscous liquid formulations. Both are beyond the scope of this review.

  1. Table 3 is too long. Divide them into 2 or more parts. – Yes, it is long. It clearly demonstrates the widespread biomedical applications of calcium orthophosphates.

Reviewer 2 Report

The study entitled “Calcium orthophosphate (CaPO4)-based bioceramics: preparation, properties, and applications” is an interesting domain of material science but was not a novel piece of work because the same work has already been published in “materials” journals. The work isn’t considered novel and it has no significance in scientific literature.

1.      This paper makes no scientific value. Because the information has already been published.

2.      The title describes the applications but the biomedical application, biological properties, and in vivo behavior of bioceramics is missing in the provided literature, which also weakens the cited literature.

3.      The link to the published literature which has a similar study is provided below.

https://www.ncbi.nlm.nih.gov/pmc/articles/PMC5452669/

4.      The published literature is from the same author; the paper even doesn’t have a different language.

5.      Yes, the topic is timely and will be of interest to the readers of the journal.  However, the manuscript is very poorly written and the ideas do not flow logically. 

6.      The review of the literature is not thorough so it does not give an adequate background about the topic.

7.      Numerous statements are made that reflect the authors’ opinion but they are portrayed as fact with no references cited. 

8.      Most of the content in the manuscript is well known by the readers of the journal.

Author Response

Dear unknown reviewer!

First, thank you very much for spending your valuable time, kind efforts and suggestions to make my manuscript even better. Second, please, find below a general description of my approach, followed by the point-to-point responses.

Yours sincerely,

Sergey V. Dorozhkin

General description:

I introduce a novel (for science) approach of continuous revising and updating of the published reviews of mine, because this is the only way to provide customers (in our case, other researchers) with the products of the highest quality. Let me mention some examples from other aspects of human beings. For instance, what is the difference between two models of cars (e.g., Mercedeses) designed in 2021 and 2022, respectively? Actually, there are almost no differences, except of minor changes in construction and design plus addition of a bit more computers and electronics. However, the more recent model is always advertised as a “new Mercedes”, if compared with the previous one. Nobody starts designing each new model of a car from the very beginning, such as each time a new type of wheels, a new type of engine, etc. (this will be both very long and very expensive) – the previously designed wheels, engine, etc. are always used plus minor improvements. Strange enough, but nobody complains on “duplications” of the major parts of the model 2021 in the model 2022 and both of them on “duplications” of the major parts taken from the models of 2020, 2019, 2018, etc. Similar is valid for software: more recent versions always include everything containing in previous ones plus some new options and features. Again, nobody rewrites the program code from the very beginning for each new version and, again, in spite of many similarities, nobody complains on “duplications”. The amount of the examples is endless because this appears to be the only approach to create the quality. Strange enough, but this simple approach is not used in writing scientific reviews, each time any new review is composed from the very beginning. As a result, according to the databases, 1200+ reviews on apatites and calcium phosphates have been already published, but only 170 of them were cited 100+ times. No catastrophe would have been happened if the half of poorly cited reviews had not been published at all. If they disappear, nobody will notice this. I am trying to improve the situation. That is why, I propose to further improve the time proved high-quality review of mine published in 2013 (173 citations according to Scopus and 159 citations according to Web of Science) to create the product of even a higher quality, which will be well cited and by means of this it will make a serious contribution to the impact factor increasing of the journal.

Reviewer 2

Comments and Suggestions for Authors

The study entitled “Calcium orthophosphate (CaPO4)-based bioceramics: preparation, properties, and applications” is an interesting domain of material science but was not a novel piece of work because the same work has already been published in “materials” journals. The work isn’t considered novel and it has no significance in scientific literature. – Thank you for your frank opinion, see General description.

  1. This paper makes no scientific value. Because the information has already been published. Correct, see General description.

  1. The title describes the applications but the biomedical application, biological properties, and in vivo behavior of bioceramics is missing in the provided literature, which also weakens the cited literature. – I am both a chemist and a materials researcher; therefore the topics of biomedical application, biological properties, and in vivo behavior of bioceramics are not the strongest parts of my background. Let a clinician to review all those topics.

  1. The link to the published literature which has a similar study is provided below.

https://www.ncbi.nlm.nih.gov/pmc/articles/PMC5452669/

Correct, see General description

  1. The published literature is from the same author; the paper even doesn’t have a different language. – In my humble opinion, it is useless to rephrase sentences (e.g., replace “sample A was studied by XRD” by “an XRD technique was used to study the properties of A) just to avoid duplications because it is a very big and hard job, which has the zero scientific sense.

  1. Yes, the topic is timely and will be of interest to the readers of the journal. However, the manuscript is very poorly written and the ideas do not flow logically. – Please, provide the details.

  1. The review of the literature is not thorough so it does not give an adequate background about the topic. – Please, provide the details.

  1. Numerous statements are made that reflect the authors’ opinion but they are portrayed as fact with no references cited. – Please, provide the details.

  1. Most of the content in the manuscript is well known by the readers of the journal. Correct, see General description.

Reviewer 3 Report

The work is done at a high level, all topics are described deeply and meaningfully. However, in the description of the application methods, please add the Plasma Electrolytic Oxidation in aqueous solution and molten salt method, its new and advanced technology for applying bioactive coatings such as hydroxyapatite.

Author Response

Dear unknown reviewer!

First, thank you very much for spending your valuable time, kind efforts and suggestions to make my manuscript even better. Second, please, find below the point-to-point responses.

Yours sincerely,

Sergey V. Dorozhkin

Reviewer 3

Comments and Suggestions for Authors

The work is done at a high level, all topics are described deeply and meaningfully. However, in the description of the application methods, please add the Plasma Electrolytic Oxidation in aqueous solution and molten salt method, its new and advanced technology for applying bioactive coatings such as hydroxyapatite. – Declined, because bioactive coatings such as hydroxyapatite is a separate subject, already reviewed by various scientists, including myself. Please, see the following review of mine, which contains a section on Plasma Electrolytic Oxidation:

S.V. Dorozhkin, Calcium orthophosphate deposits: preparation, properties and biomedical applications. Materials Science and Engineering C 2015, 55, 272-326.

http://www.sciencedirect.com/science/article/pii/S0928493115300837

Round 2

Reviewer 2 Report

I have no comments to the author, However, I have submitted specific comments to editor.